# Adhesive anti-fibrotic interfaces on diverse organs

Jingjing Wu[1], Jue Deng[1], Georgios Theocharidis[2], Tiffany L. Sarrafian[3], Leigh G. Griffiths[4], Roderick T. Bronson[5], Aristidis Veves[2], Jianzhu Chen[6], Hyunwoo Yuk[1,8 ✉] & Xuanhe Zhao[1,7 ✉]

Implanted biomaterials and devices face compromised functionality and efficacy in the long term owing to foreign body reactions and subsequent formation of fibrous capsules at the implant–tissue interfaces[1–4]. Here we demonstrate that an adhesive implant–tissue interface can mitigate fibrous capsule formation in diverse animal models, including rats, mice, humanized mice and pigs, by reducing the level of infiltration of inflammatory cells into the adhesive implant–tissue interface compared to the non-adhesive implant–tissue interface. Histological analysis shows that the adhesive implant–tissue interface does not form observable fibrous capsules on diverse organs, including the abdominal wall, colon, stomach, lung and heart, over 12 weeks in vivo. In vitro protein adsorption, multiplex Luminex assays, quantitative PCR, immunofluorescence analysis and RNA sequencing are additionally carried out to validate the hypothesis. We further demonstrate long-term bidirectional electrical communication enabled by implantable electrodes with an adhesive interface over 12 weeks in a rat model in vivo. These findings may offer a promising strategy for long-term anti-fibrotic implant–tissue interfaces.

Foreign body reactions to implants are among the most critical challenges that undermine the long-term functionality and reliability of biomaterials and devices in vivo[1–4]. In particular, the formation of a fibrous capsule between the implant and the target tissue, as a result of foreign body reactions, can substantially compromise the implant's efficacy because the fibrous capsule acts as a barrier to mechanical, electrical, chemical or optical communications[4–11] (Fig. 1a,b). To alleviate the formation of the fibrous capsule at the implant–tissue interface, various approaches have been developed, including drug-eluting coatings[12], hydrophilic[13] or zwitterionic polymer coatings[14–16], active surfaces[17,18] and controlling the stiffness[19] and/or size[20,21] of the implants. However, despite recent advances, the mitigation of fibrous capsule formation for implanted biomaterials and devices remains an ongoing challenge in the field[5,22], highlighting the importance of developing new solutions and strategies.

Here we demonstrate that an adhesive interface can not only provide mechanical integration of the implant with the target tissue but also prevent the formation of observable fibrous capsules at the implant–tissue interface (Fig. 1c,d). We reason that the conformal interfacial integration between the adhesive implant and the tissue surface can reduce the level of infiltration of inflammatory cells (for example, neutrophils, monocytes, macrophages) into the adhesive implant–tissue interface, resulting in a decreased level of collagen deposition and a reduced level of fibrous capsule formation in the long term (Fig. 1d). By contrast, conventional non-adhesive implants usually do not form conformal integration with the tissue surfaces and attract the infiltration of inflammatory cells into the non-adhesive implant–tissue interfaces. Subsequently, fibrous capsules form on the non-adhesive implant–tissue interfaces (Fig. 1b).

To test our hypothesis, we prepared an adhesive implant consisting of a mock device (polyurethane) and an adhesive layer[23,24] composed of interpenetrating networks between the covalently crosslinked poly(acrylic acid) N-hydroxysuccinimide ester and physically crosslinked poly(vinyl alcohol) (Fig. 1c). The adhesive layer provides highly conformal and stable integration of the implant with wet tissues[23–25] (Supplementary Fig. 1). We further prepared a non-adhesive implant by fully swelling the same mock device and adhesive layer in a phosphate-buffered saline bath before implantation (see Methods for the preparation of the non-adhesive implant). By swelling the implant in phosphate-buffered saline, we removed its adhesive property[26] while keeping its chemical composition identical.

Both adhesive and non-adhesive implants were implanted on the surfaces of diverse organs, including the abdominal wall, colon, stomach, lung and heart, using rat models in vivo for up to 84 days (Fig. 1e–i). Note that the non-adhesive implant was sutured onto the organ surfaces. Macroscopic observations showed that both adhesive and non-adhesive implants remained stable at the implantation site on the organ surfaces (Extended Data Fig. 1b,c). To analyse the foreign body reaction and fibrous capsule formation for the adhesive and non-adhesive implants, we carried out histological analysis of the native tissue, adhesive implant and non-adhesive implant for various organs (Extended Data Fig. 1a).

Histological evaluation by a blinded pathologist indicates that the adhesive implant forms conformal integration with the organ surface

[1]Department of Mechanical Engineering, Massachusetts Institute of Technology, Cambridge, MA, USA. [2]Joslin-Beth Israel Deaconess Foot Center and The Rongxiang Xu, MD, Center for Regenerative Therapeutics, Beth Israel Deaconess Medical Center, Harvard Medical School, Boston, MA, USA. [3]Department of Thoracic Surgery, Mayo Clinic, Rochester, MN, USA. [4]Department of Cardiovascular Medicine, Mayo Clinic, Rochester, MN, USA. [5]Department of Immunology, Harvard Medical School, Boston, MA, USA. [6]Koch Institute for Integrative Cancer Research and Department of Biology, Massachusetts Institute of Technology, Cambridge, MA, USA. [7]Department of Civil and Environmental Engineering, Massachusetts Institute of Technology, Cambridge, MA, USA. [8]Present address: SanaHeal, Cambridge, MA, USA. ✉e-mail: hyunwooyuk@sanaheal.com; zhaox@mit.edu

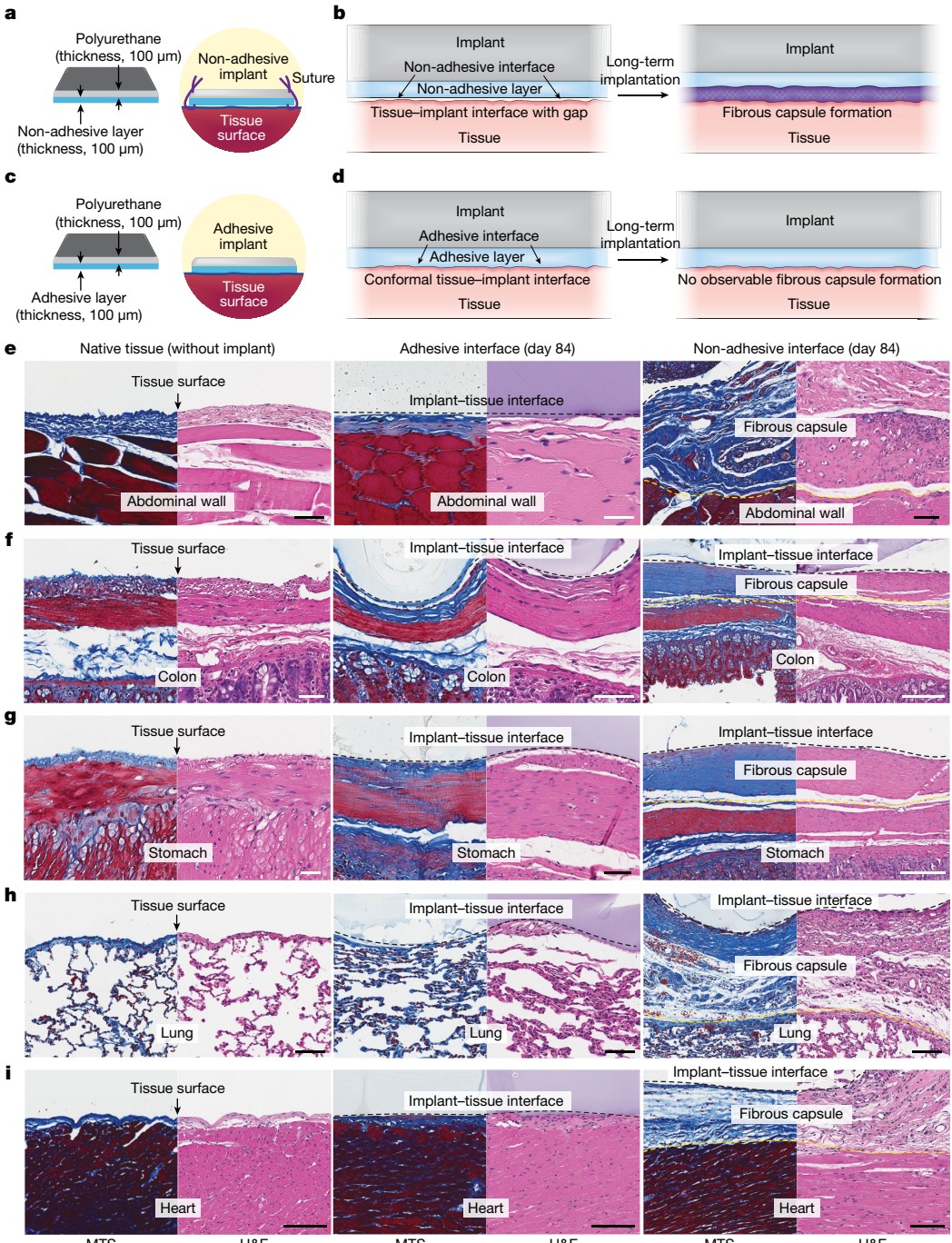

**Fig. 1 | Adhesive anti-fibrotic interfaces. a,b**, Schematic illustrations of a non-adhesive implant consisting of a mock device (polyurethane) and a non-adhesive layer (**a**) and long-term in vivo implantation with fibrous capsule formation at the implant–tissue interface (**b**). **c,d**, Schematic illustrations of an adhesive implant consisting of the mock device (polyurethane) and an adhesive layer (**c**) and long-term in vivo implantation without observable fibrous capsule formation at the implant–tissue interface (**d**). **e–i**, Representative histology images stained with Masson's trichrome (MTS) and haematoxylin and

eosin (H&E) for native tissue (left), the adhesive implant (middle) and the non-adhesive implant (right) collected on day 84 post-implantation on the abdominal wall (**e**), colon (**f**), stomach (**g**), lung (**h**) and heart (**i**). Black and yellow dashed lines in the images indicate the implant–tissue interface and the fibrous capsule–tissue interface, respectively. The experiment in **e–i** was repeated independently (*n* = 4 per group) with similar results. Scale bars, 50 µm (**e–g**, left and middle; **h**), 100 µm (**e**, right; **i**), 200 µm (**f**, right), 150 µm (**g**, right).

and shows no observable formation of the fibrous capsule up to 84 days post-implantation for diverse organs, including the abdominal wall, colon, stomach, lung and heart (Fig. 1e–i, Extended Data Fig. 2 and Supplementary Fig. 2). Furthermore, a transmission electron micrograph of the adhesive implant–tissue interface shows that the adhesive layer maintains highly conformal integration with the collagenous layer of the mesothelium on a subcellular scale on day 28

post-implantation (Extended Data Fig. 3). By contrast, the non-adhesive implant undergoes substantial formation of the fibrous capsule at the implant–tissue interface for all organs, consistent with the foreign body reaction to the mock device alone (Fig. 1 and Supplementary Figs. 2 and 3). Similarly, the mock device–cavity interface of the adhesive implant undergoes fibrous capsule formation (the top of the implant in Extended Data Fig. 2).

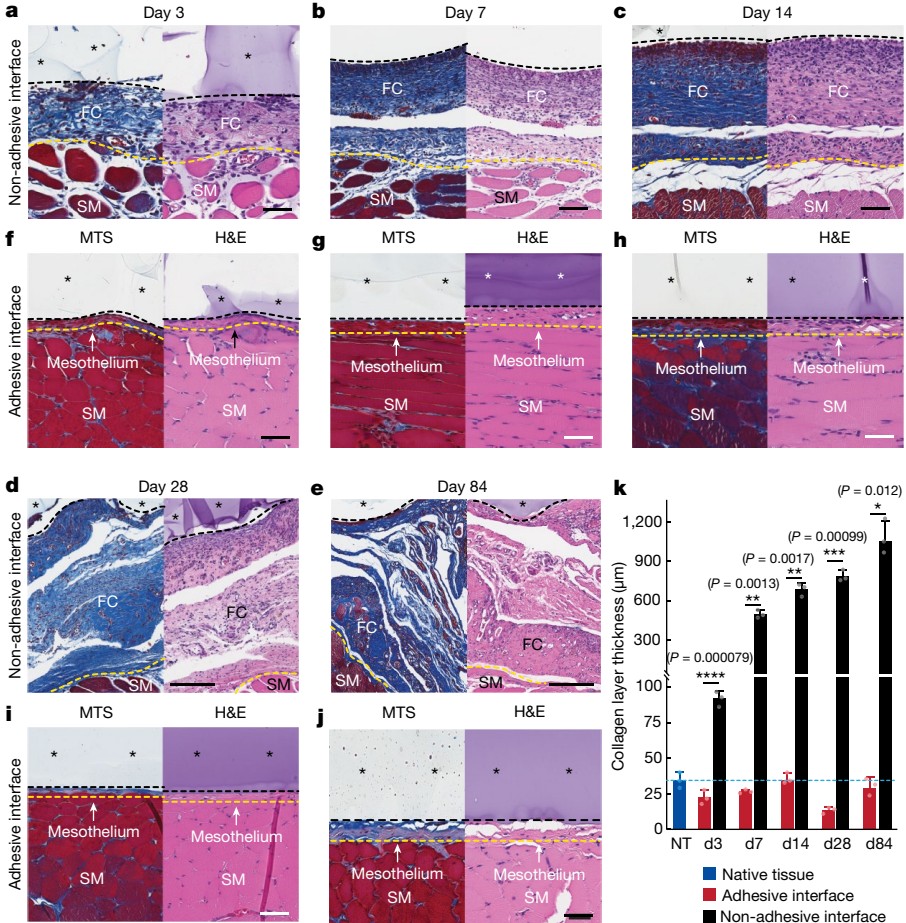

**Fig. 2 | Histology analysis of the adhesive and non-adhesive implant–tissue interfaces at different time points. a–e**, Representative histology images stained with Masson's trichrome (left) and haematoxylin and eosin (right) of the non-adhesive implant collected on day 3 (**a**), day 7 (**b**), day 14 (**c**), day 28 (**d**) and day 84 (**e**) post-implantation on the abdominal wall. **f–j**, Representative histology images stained with Masson's trichrome (left) and haematoxylin and eosin (right) of the adhesive implant collected on day 3 (**f**), day 7 (**g**), day 14 (**h**), day 28 (**i**) and day 84 (**j**) post-implantation on the abdominal wall. Asterisks in images indicate the implant; black dashed lines in images indicate the implant–tissue interface; yellow dashed lines in images indicate the mesothelium–fibrous capsule (non-adhesive implant) or the mesothelium–skeletal muscle (adhesive implant) interface. SM, skeletal muscle; FC, fibrous capsule. **k**, Collagen layer thickness at the implant–tissue interface measured at different time points post-implantation. The blue dashed line indicates the average collagen layer thickness of the native tissue (NT). d, day. Values in **k** represent the mean and the standard deviation (*n* = 3 implants; independent biological replicates). Statistical significance and *P* values were determined by two-sided unpaired *t*-tests; \**P* < 0.05; \*\**P* ≤ 0.01; \*\*\**P* ≤ 0.001; \*\*\*\**P* < 0.0001. Scale bars, 50 μm (**a**,**f–j**), 100 μm (**b**,**c**), 200 μm (**d**,**e**).

To investigate the potential influence of suture-induced tissue damage, sutures were introduced to the corners of the adhesive implant, similar to those used with the non-adhesive implant (Extended Data Fig. 4a). The histological analysis shows that the suture point exhibits the formation of fibrosis (Extended Data Fig. 4b,c), but the intact adhesive implant–tissue interface demonstrates no observable formation of the fibrotic capsule (Extended Data Fig. 4b,d). Collectively, these data further confirm that the adhesive interface is required to prevent the observable formation of the fibrous capsule.

To investigate the effect of adhesive interfaces with varying compositions and properties, we replaced the poly(vinyl alcohol)-based adhesive interface with a chitosan-based adhesive interface[23] (see Methods for the preparation of the chitosan-based adhesive interface). Compared to the poly(vinyl alcohol)-based adhesive interface, the chitosan-based adhesive interface offers a different composition and Young's modulus, yet it demonstrates comparable adhesion performance (Extended Data Fig. 5a–d). Histological analysis shows that the chitosan-based adhesive interface exhibits no observable formation of the fibrous capsule on day 14 post-implantation (Extended Data Fig. 5e,f). Notably, the implants adhered to the abdominal wall surface

using commercially available tissue adhesives including Coseal and Tisseel show the substantial formation of the fibrous capsule on day 14 post-implantation (Extended Data Fig. 6). This may be attributed to unstable long-term adhesion of the commercially available tissue adhesives with the tissue surface in vivo[27,28].

To assess the foreign body reaction and fibrous capsule formation over time, we conducted histological analyses for the adhesive and non-adhesive implants on the abdominal wall on days 3, 7, 14, 28 and 84 post-implantation (Fig. 2a–j). The collagen layer thickness at the implant–tissue interface remains comparable to that of the native tissue (that is, the mesothelium thickness) for the adhesive implant at all time points (Fig. 2k). By contrast, the collagen layer thickness at the non-adhesive implant–tissue interface increases over time owing to the formation of the fibrous capsule and is significantly thicker than that of both the native tissue and the adhesive implant at all time points (Fig. 2k).

To further investigate our hypothesis, we carried out a set of characterizations for key participants of the foreign body reaction, including in vitro protein adsorption assays, immunofluorescence analysis, quantitative PCR (qPCR), Luminex quantification and RNA-sequencing

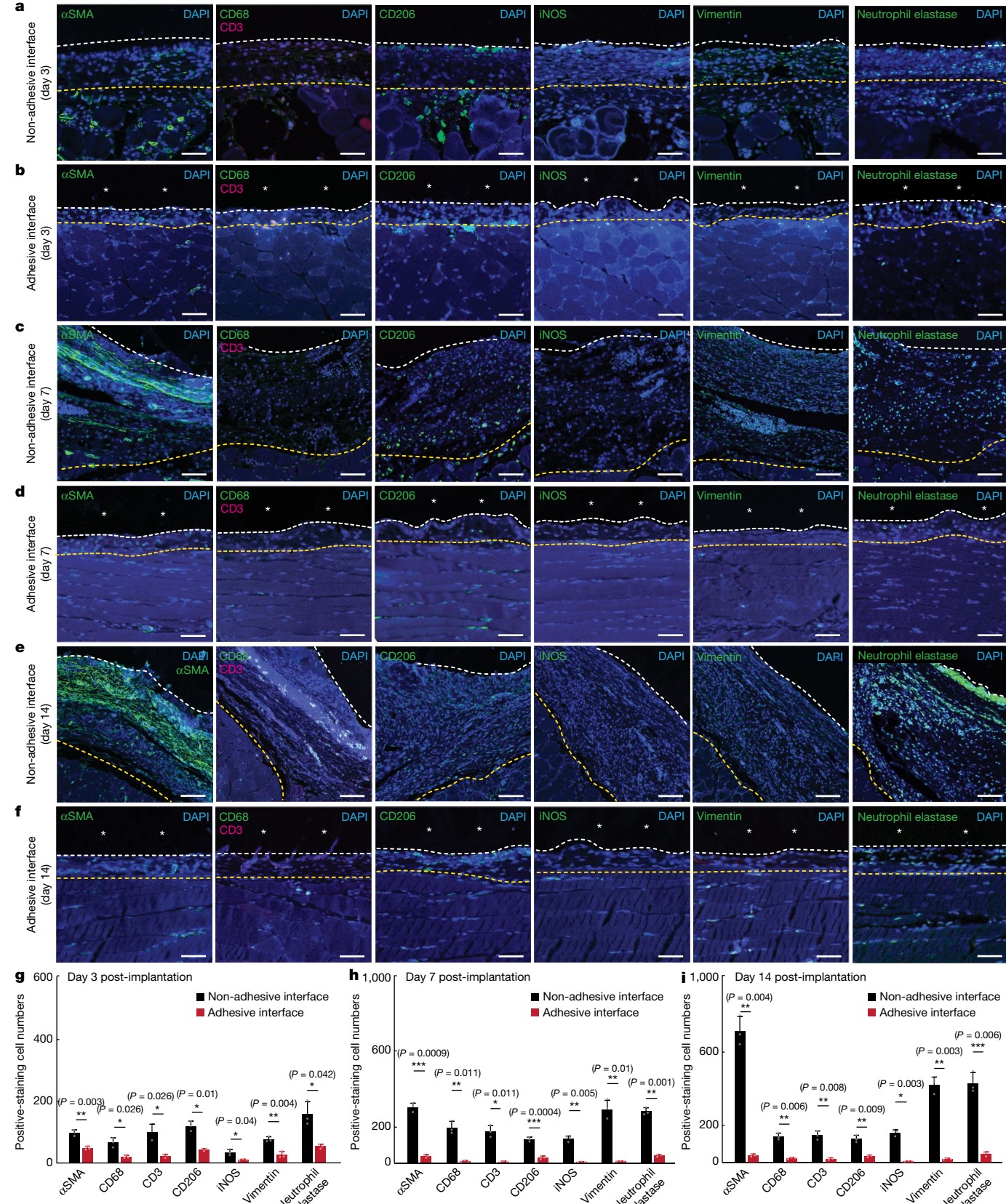

**Fig. 3 |** See next page for caption.

analysis. A protein adsorption assay with fluorescently labelled albumin and fibrinogen was carried out to evaluate the adhesion of proteins at the implant–tissue interface during the initial stage of the foreign body reaction[29,30] (Supplementary Fig. 4). After 30 min of co-culture in the protein solution, the adhesive implant–substrate interface showed a significantly lower level of protein adsorption compared to that of the

**Fig. 3 | Immunofluorescence analysis of the adhesive and non-adhesive implant–tissue interfaces at different time points. a,c,e,** Representative immunofluorescence images of the non-adhesive implant collected on day 3 (**a**), day 7 (**c**) and day 14 (**e**) post-implantation on the abdominal wall. **b,d,f,** Representative immunofluorescence images of the adhesive implant collected on day 3 (**b**), day 7 (**d**) and day 14 (**f**) post-implantation on the abdominal wall. In immunofluorescence images, cell nuclei are stained with 4′,6-diamidino-2-phenylindole (DAPI, blue); green fluorescence corresponds to the staining of fibroblasts (αSMA), neutrophils (neutrophil elastase) and macrophages (CD68, vimentin, CD206, iNOS); red fluorescence corresponds to the staining of T cells (CD3). Asterisks in images indicate the implant; white dashed lines in images indicate the implant–tissue interface; yellow dashed lines in images indicate either the mesothelium–fibrous capsule interface (non-adhesive implant) or the mesothelium–skeletal muscle interface (adhesive implant). **g–i,** Quantification of cell numbers in the collagenous layer at the implant–tissue interface over a representative width of 500 μm from the immunofluorescence images on day 3 (**g**), day 7 (**h**) and day 14 (**i**) post-implantation. Values in **g–i** represent the mean and the standard deviation (*n* = 3 implants; independent biological replicates). Statistical significance and *P* values were determined by two-sided unpaired *t*-tests; NS, not significant; **P* < 0.05; ***P* ≤ 0.01; ****P* ≤ 0.001. Scale bars, 20 μm (**a,b,d,f**), 40 μm (**c,e**).

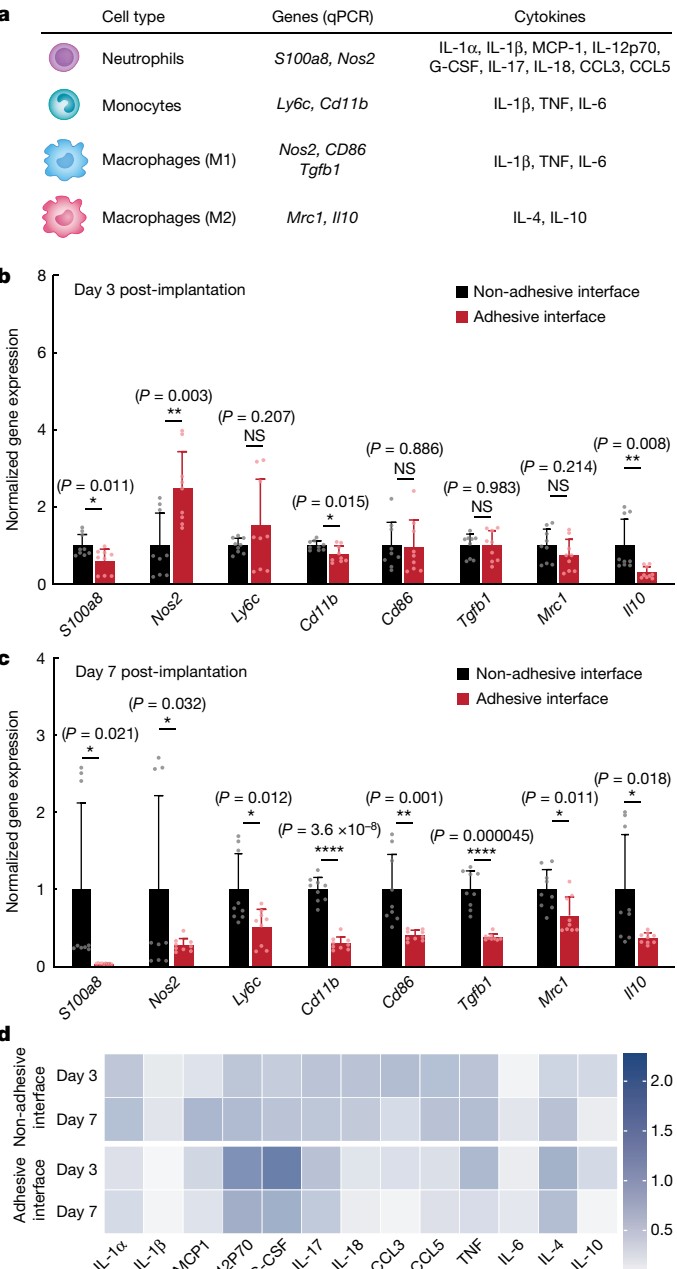

**Fig. 4 | qPCR and Luminex analysis of the adhesive and non-adhesive implant–tissue interfaces. a,** Genes and cytokines relevant to each cell type in the qPCR and Luminex studies. **b,c,** Normalized gene expression of immune-cell-related markers for the non-adhesive and the adhesive implant–tissue interface collected on day 3 (**b**) and day 7 (**c**) post-implantation on the abdominal wall. **d,** Heat map of immune-cell-related cytokines measured with Luminex assay of the non-adhesive and the adhesive implant–tissue interfaces collected on days 3 and 7 post-implantation on the abdominal wall. Values in **b,c** represent the mean and the standard deviation (*n* = 9 implants; independent biological replicates). Statistical significance and *P* values were determined by two-sided unpaired *t*-tests; NS, not significant; **P* < 0.05; ***P* ≤ 0.01; *****P* < 0.0001.

non-adhesive implant–substrate interface (*P* < 0.0001) for both fluorescently labelled albumin and fibrinogen (Supplementary Fig. 4g,h), demonstrating the adhesive interface's capability to prevent protein adsorption.

To investigate the infiltration of immune cells into the implant–tissue interface, we carried out immunofluorescence staining for fibroblasts (αSMA), neutrophils (neutrophil elastase), macrophages (CD68 for pan-macrophages; iNOS and vimentin for pro-inflammatory macrophages; CD206 for anti-inflammatory macrophages) and T cells (CD3) on days 3, 7 and 14 post-implantation (Fig. 3a–f). Quantification of cell numbers in the collagenous layer at the implant–tissue interface over a representative width of 500 μm from the immunofluorescence images shows significantly fewer fibroblasts, neutrophils, macrophages and T cells at the adhesive implant–tissue interface than at the non-adhesive implant–tissue interface at all time points (Fig. 3g–i).

To further delineate the immune response at the implant–tissue interface, we profile immune-cell-related genes and cytokines using qPCR analysis and Luminex quantification, respectively (Fig. 4). On day 3 post-implantation, whereas the levels of most select immune gene transcripts are similar or significantly lower in the adhesive compared to the non-adhesive implant–tissue interface, the level of *Nos2* expression is significantly higher in the adhesive than in the non-adhesive implant–tissue interface (Fig. 4b). The higher level of *Nos2* expression is in agreement with the higher levels of inflammatory cytokines (G-CSF, IL-12p70) in the adhesive than in the non-adhesive implant–tissue interface on day 3 post-implantation (Fig. 4d and Supplementary Table 1).

To investigate the source of *Nos2* expression on day 3 post-implantation, we carried out double immunofluorescence staining for iNOS and neutrophil elastase and for iNOS and CD68 (Extended Data Fig. 7). The immunofluorescence staining of the adhesive implant–tissue interface reveals a significantly higher number of iNOS⁺ neutrophils than iNOS⁺ macrophages on day 3 post-implantation (*P* ≤ 0.01; Extended Data Fig. 7b). By contrast, the non-adhesive implant–tissue interface has similar numbers of iNOS⁺ neutrophils and iNOS⁺ macrophages on day 3 post-implantation (*P* = 0.82; Extended Data Fig. 7d). This result indicates that the adhesive implant–tissue interface favours an iNOS-producing neutrophil subset on day 3 post-implantation[31].

By day 7 post-implantation, the adhesive implant–tissue interface exhibits a significantly lower expression level of all immune-cell-related genes, including *Nos2*, compared to the non-adhesive implant–tissue interface (Fig. 4c), consistent with the reduction in the level of inflammatory cytokines in the adhesive implant–tissue interface

on day 7 post-implantation compared to day 3 post-implantation (Fig. 4d and Supplementary Table 1). Thus, the adhesive implant–tissue interface seems to induce a more robust pro-inflammatory neutrophil response than that of the non-adhesive implant–tissue interface on day 3 post-implantation, which is rapidly resolved by day 7 post-implantation.

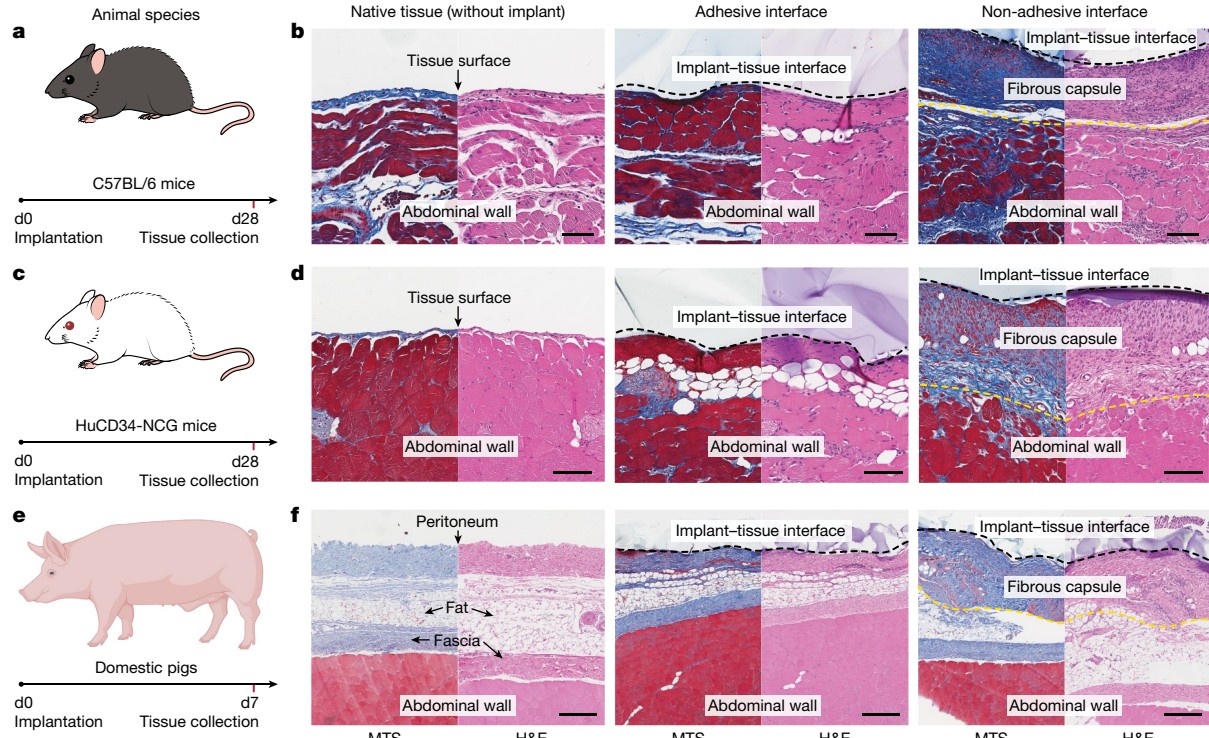

**Fig. 5 | Adhesive anti-fibrotic interfaces in diverse animal models.** **a,c,e,** Schematic illustrations for the study design in C57BL/6 mice (**a**), HuCD34-NCG humanized mice (**c**) and pigs (**e**). Implants are placed on the abdominal wall of the animals. **b,d,f,** Representative histology images stained with Masson's trichrome and haematoxylin and eosin for native tissue (left), the adhesive implant (middle) and the non-adhesive implant (right) collected on day 28 post-implantation in C57BL/6 mice (**b**) and HuCD34-NCG humanized mice (**d**), and on day 7 post-implantation in pigs (**f**). Black dashed lines in images indicate the implant–tissue interface; yellow dashed lines in images indicate the fibrous capsule–tissue interface. The experiment in **b,d,f** was repeated independently (n = 6 per group for C57BL/6 mice; n = 5 per group for HuCD34-NCG mice; n = 4 per group for pigs) with similar results. Scale bars, 100 μm (**b,d**), 300 μm (**f**). The graphic of the pig in **e** was created with BioRender.com.

Next we carried out bulk RNA sequencing of implant–abdominal wall interfaces for both adhesive and non-adhesive implants on days 3 and 14 post-implantation to further investigate gene expression differences (Extended Data Fig. 8). Principal component analysis shows separate clustering of samples for the non-adhesive and adhesive implant–tissue interfaces at each time point, indicating distinct transcriptomic profiles (Extended Data Fig. 8a,d). Differential gene expression analysis of the adhesive compared to the non-adhesive implant–tissue interface reveals 40 downregulated and 33 upregulated genes on day 3 post-implantation (Extended Data Figs. 8b and 9a). On day 14 post-implantation, 357 genes are downregulated and 156 genes are upregulated (Extended Data Figs. 8e and 9b) in the adhesive implant–tissue interface compared to the non-adhesive implant–tissue interface. On day 3 post-implantation, regulation of interferon production and striated muscle tissue development are enriched in the non-adhesive implant–tissue interface, indicating inflammatory and fibrosis processes, whereas cell proliferation and growth processes are enriched in the adhesive implant–tissue interface (Extended Data Fig. 8c). On day 14 post-implantation, fibrosis-associated processes are highly enriched in the non-adhesive implant–tissue interface, such as muscle cell differentiation, myofibril assembly and muscle structure development, whereas vasculature formation, neurogenesis and proliferation are enriched in the adhesive implant–tissue interface (Extended Data Fig. 8f). These results again indicate reduced inflammatory response and rapid resolution of inflammation in the adhesive implant–tissue interface compared to the non-adhesive implant–tissue interface.

To test our hypothesis in diverse animal models, we implanted the adhesive and non-adhesive implants on the abdominal wall surface of immunocompetent C57BL/6 mice and HuCD34-NCG humanized mice (Fig. 5a,c). Note that immunocompetent C57BL/6 mice are known to produce fibrosis and foreign body reactions similar to those observed in human patients[32], and HuCD34-NCG humanized mice provide human-like immune responses[33]. Histological analysis shows that the adhesive implant–tissue interface exhibits no observable formation of the fibrous capsule, comparable to the native tissue on day 28 post-implantation in both C57BL/6 (Fig. 5b) and HuCD34-NCG (Fig. 5d) mouse models. By contrast, the non-adhesive implant–tissue interface shows substantial formation of the fibrous capsule in both models (Fig. 5b,d).

To further test our hypothesis in human-scale anatomy, we implanted the adhesive and non-adhesive implants in porcine models (Fig. 5e and Supplementary Fig. 5). Macroscopic observations demonstrate that the adhesive implant maintains stable integration with the surface of the porcine abdominal wall and small intestine on day 7 post-implantation in vivo (Extended Data Fig. 10). Histological analysis shows that the adhesive implant forms conformal integration with the tissue surface without observable formation of the fibrous capsule on the implant–tissue interface on day 7 post-implantation for both the abdominal wall (Fig. 5e) and small intestine (Extended Data Fig. 10a). By contrast, the non-adhesive implant–tissue interface exhibits substantial formation of the fibrous capsule (Fig. 5f and Extended Data Fig. 10b), in agreement with the observations in the rodent models.

To explore the potential utility of the adhesive anti-fibrotic interfaces, we demonstrated long-term in vivo electrophysiological recording and stimulation enabled by the implantable electrodes with the adhesive interface in a rat model for 84 days (Fig. 6). For continuous in vivo monitoring and modulation of the electrocardiogram, electrodes with either the adhesive or non-adhesive interface were implanted on the epicardial surface of animals for electrophysiological recording and

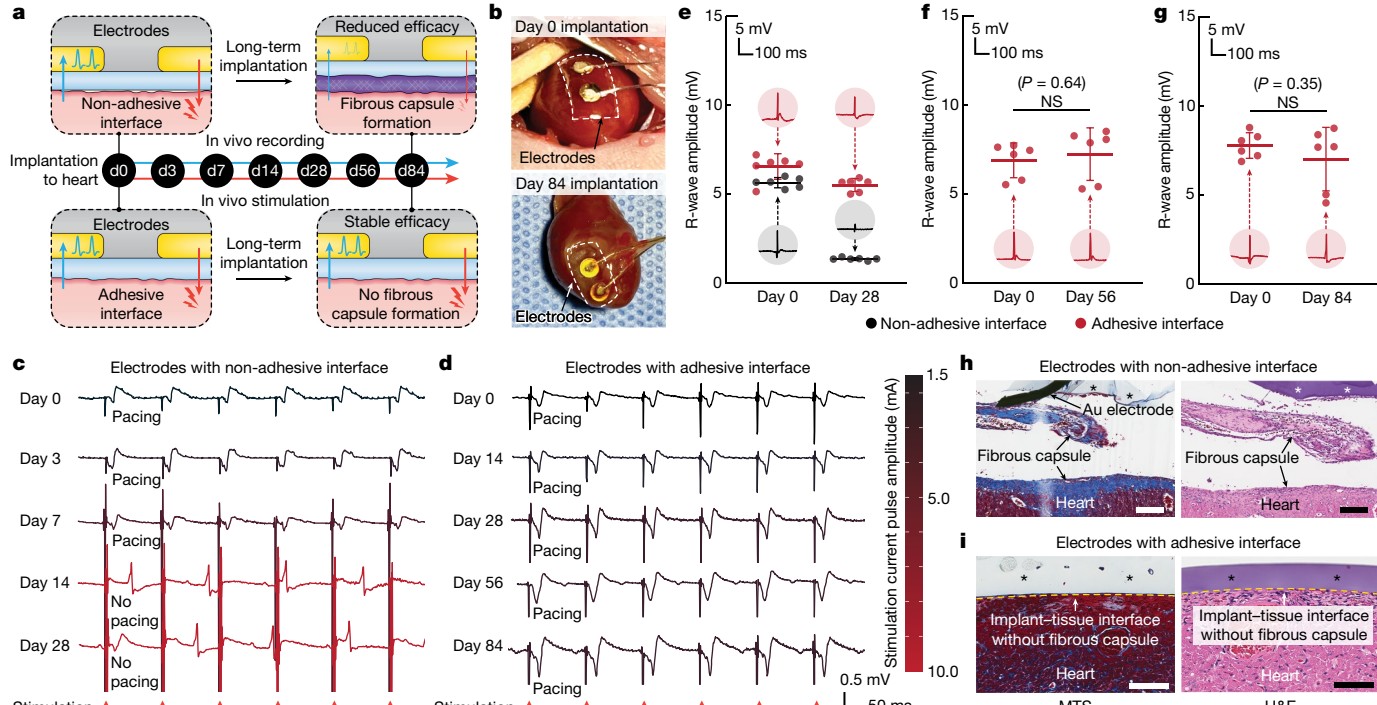

**Fig. 6 | Long-term in vivo bidirectional electrical communication through the adhesive anti-fibrotic interfaces. a**, Schematic illustrations for the in vivo electrophysiological recording and stimulation through implanted electrodes with the non-adhesive or the adhesive implant–tissue interface. **b**, Photographs of the heart collected on days 0 and 84 post-implantation for electrodes with the adhesive interface. White dashed lines in photographs indicate the boundary of implants. **c**, Representative epicardial electrocardiograms after stimulation through implanted electrodes with the non-adhesive implant–tissue interface on days 0, 3, 7, 14 and 28 post-implantation on a rat heart. **d**, Representative epicardial electrocardiograms after stimulation through implanted electrodes with the adhesive implant–tissue interface on days 0, 14, 28, 56 and 84 post-implantation on a rat heart. **e**–**g**, Recorded R-wave amplitude

through implanted electrodes with the non-adhesive (black) and the adhesive (red) implant–tissue interfaces on day 28 (**e**), day 56 (**f**) and day 84 (**g**) post-implantation on a rat heart. Inset plots show representative recorded waveforms. **h,i**, Representative histology images stained with Masson's trichrome (left) and haematoxylin and eosin (right) of the electrodes with the non-adhesive (**h**) and the adhesive (**i**) implant collected on day 28 post-implantation on a rat heart. Asterisks in images indicate the implant; yellow dashed lines in images indicate the implant–tissue interface. Values in **e**–**g** represent the mean and the standard deviation (n = 6 animals; independent biological replicates). The experiment in **h,i** was repeated independently (n = 6 per group) with similar results. Statistical significance and P values were determined by two-sided unpaired t-tests; NS, not significant. Scale bars, 200 µm (**h**), 100 µm (**i**).

stimulation on days 0, 3, 7, 14, 28, 56 and 84 post-implantation (Fig. 6a and Supplementary Fig. 6). Macroscopic observations showed that the electrodes with the adhesive interface maintained stable integration with the heart after 84 days of implantation in vivo (Fig. 6b). The amplitude of the R wave recorded by the electrodes with the adhesive interface was consistently maintained throughout the study duration (84 days; Fig. 6e–g), whereas the R-wave amplitude recorded by the electrodes with the non-adhesive interface exhibited a substantial decrease over time (Fig. 6e). For electrophysical stimulation by the electrodes with the non-adhesive interface, the minimal stimulation current pulse amplitude needed to successfully pace the heart gradually increased until day 7 post-implantation and eventually failed to pace the heart on day 14 post-implantation (Fig. 6c). By contrast, the electrodes with the adhesive interface exhibited a consistent minimal stimulation current pulse amplitude for pacing and successfully maintained the capability to pace the heart for the duration of the study (84 days; Fig. 6d). These results are consistent with the histological findings from the tissues collected on day 28 post-implantation, for which the electrodes with the non-adhesive interface showed encapsulation and physical separation from the epicardial surface by a thick fibrous capsule (Fig. 6h). By contrast, the electrodes with the adhesive interface showed conformal contact with the epicardial surface without observable formation of the fibrous capsule (Fig. 6i).

In this study, we demonstrated that the adhesive interface can not only provide conformal mechanical integration of the implant to the target tissue but also effectively mitigate the formation of the fibrous

capsule on the adhesive implant–tissue interface by reducing the level of infiltration of inflammatory cells. The current work provides a promising strategy for long-term anti-fibrotic implant–tissue interfaces and offers valuable insights into implant–tissue interactions for future studies.

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

## Methods

### Preparation of adhesive implants

The adhesive layer of the adhesive implant was prepared using a previously reported method[23,24]. To prepare an adhesive stock solution, 35% w/w acrylic acid, 7% w/w poly(vinyl alcohol) (PVA; $M_w$ = 146,000–186,000, 99+% hydrolysed), 0.2% w/w α-ketoglutaric acid and 0.05% w/w $N,N'$-methylenebisacrylamide were added into nitrogen-purged deionized water. Next, 30 mg of acrylic acid $N$-hydroxysuccinimide ester was dissolved in each 1 ml of the above stock solution to prepare the adhesive precursor solution. The chitosan-based adhesive layer was prepared by replacing PVA with 2% w/w chitosan (Mw = 250–300 kDa, degree of deacetylation > 90%; ChitoLytic). The precursor solution was poured onto a glass mould with a spacer (100-µm thickness) and placed in a UV chamber (354 nm, 12 W power) for 30 min to prepare the adhesive hydrogel. The adhesive hydrogel was dried thoroughly under airflow and a vacuum desiccator to prepare the dry adhesive layer. A mock device of the adhesive implant was introduced by spin-coating a polyurethane resin (HydroThane, AdvanSource Biomaterials) onto the dry adhesive layer.

### Preparation of non-adhesive implants

To prepare the non-adhesive implant, the adhesive implant was immersed in a sterile 1× phosphate-buffered saline (PBS; pH 7.4, 144 mg $l^{-1}$ potassium phosphate monobasic, 9,000 mg $l^{-1}$ sodium chloride and 795 mg $l^{-1}$ sodium phosphate dibasic) bath at room temperature overnight. During this process, the adhesive layer of the implant reached the equilibrium swollen state and became non-adhesive by losing the capability to form physical (hydrogen bonds) and covalent (amide bonds) crosslinking with tissues[26].

### Preparation of implantable electrodes

To prepare the implantable electrodes, gold electrodes (thickness, 50 µm) were integrated between the polyurethane layer (thickness, 100 µm) and the adhesive or non-adhesive layer (thickness, 100 µm; Supplementary Fig. 6a). The surface of the gold electrode was treated with oxygen plasma for 3 min (30 W power, Harrick Plasma) to activate the surface functionalization, followed by immersion in cysteamine hydrochloride solution (50 mM in deionized water) for 1 h at room temperature. After the functionalization, the gold electrode was thoroughly washed with deionized water and dried with nitrogen flow. The functionalized gold electrode was cut into 2-mm-diameter circles and placed on the adhesive hydrogel (two electrodes per implant). An electrode lead wire (AS633, Cooner Wire) was connected to the gold electrodes and the polyurethane insulation layer (HydroThane, AdvanSource Biomaterials) was introduced to the gold electrodes. The assembled implant was thoroughly dried under airflow and in a vacuum desiccator to prepare the adhesive implantable electrodes. To prepare the non-adhesive implantable electrodes, the adhesive implantable electrodes were immersed in a sterile PBS bath overnight. All samples were prepared in an aseptic manner and were further disinfected under UV for 1 h before use.

### Mechanical characterization

Either the chitosan-based adhesive implant or the PVA-based adhesive implant was applied to ex vivo porcine skin with a gentle pressure for 5 s. Interfacial toughness was measured on the basis of the T-peel test (ASTM F2256). Shear strength was measured on the basis of the lap-shear test (ASTM F2255). Tensile strength was measured on the basis of the tensile test (ASTM F2258). All tests were conducted using a mechanical testing machine (2.5-kN load cell, Zwick/Roell Z2.5). Aluminium fixtures were applied using cyanoacrylate glue to provide grips for tensile tests. All mechanical characterizations were carried out three times using independently prepared samples.

### In vitro protein adsorption assay

A gelatin hydrogel (10% w/v, 300 g Bloom, Sigma-Aldrich) was used as the substrate for in vitro protein adsorption assay. The adhesive and non-adhesive implants were cut into 5-mm-diameter circles by using a biopsy punch and placed on the gelatin hydrogel. The samples were then incubated in a solution with 5 mg $ml^{-1}$ fluorescently tagged albumin (A13101, Thermo Fisher) or fibrinogen (F13191, Thermo Fisher) for 30 min. After the incubation, the samples were washed three times with fresh PBS to remove unadhered proteins. The samples were imaged using a confocal microscope (SP8, Leica), with the confocal plane set at the gelatin hydrogel–implant interface under a pitch model with excitation and emission at 495 nm and 515 nm (for albumin) and 495 nm and 635 nm (for fibrinogen). The relative fluorescence intensity of absorbed proteins was calculated by using ImageJ (version 2.1.0).

### In vivo intraperitoneal implantation in rat model

All animal studies on rats were approved by the MIT Committee on Animal Care, and all surgical procedures and postoperative care were supervised by the MIT Division of Comparative Medicine (DCM) veterinary staff.

Sprague Dawley rats (female and male, 225 to 250 g, 12 weeks, Charles River Laboratories) were used for all in vivo rat studies. Before implantation, all samples were prepared using aseptic techniques and were further disinfected for 1 h under UV light. For in vivo intraperitoneal implantation, the animals were anaesthetized using isoflurane (2 to 3% isoflurane in oxygen) in an anaesthetizing chamber before the surgery, and anaesthaesia was maintained using a nose cone throughout the surgery. Abdominal hair was removed, and the animals were placed on a heating pad during the surgery. The abdominal wall, colon or stomach was exposed by means of a laparotomy. The adhesive implant (10 mm in width and 10 mm in length) was applied to the abdominal wall ($n$ = 4 per time point), colon ($n$ = 4) or stomach ($n$ = 4) surface by gently pressing with a surgical spatula or fingertip. The non-adhesive implant (10 mm in width and 10 mm in length) was implanted on the abdominal wall ($n$ = 4 per time point), colon ($n$ = 4) or stomach ($n$ = 4) surface using sutures at the corners of the samples (8-0 Prolene, Ethicon). For commercially available tissue adhesives, 0.5 ml of Coseal ($n$ = 6) or Tisseel ($n$ = 6) was used to adhere the non-adhesive implant (10 mm in width and 10 mm in length) to the abdominal wall surface. For the adhesive implant with sutures, the adhesive implant (10 mm in width and 10 mm in length) was applied to the abdominal wall surface ($n$ = 6), and sutures (8-0 Prolene, Ethicon) were used at the corners of the samples[24]. The abdominal wall muscle and skin incisions were closed with sutures (4-0 Vicryl, Ethicon). On days 3, 7, 14, 28 and 84 post-implantation, the animals were euthanized using $CO_2$ inhalation. Abdominal wall, colon or stomach tissues of interest were excised and fixed in 10% formalin for 24 h for histological and immunofluorescence analysis. All animals in the study survived and were kept in normal health conditions on the basis of daily monitoring by the MIT DCM veterinarian staff.

### In vivo intrathoracic implantation in rat model

For in vivo intrathoracic implantation, the animals were anaesthetized using isoflurane (2 to 3% isoflurane in oxygen) in an anaesthetizing chamber before the surgery, and anaesthesia was maintained using a nose cone throughout the surgery. Chest hair was removed, and endotracheal intubation was carried out, connecting the animals to a mechanical ventilator (RoVent, Kent Scientific). The animals were placed on a heating pad for the duration of the surgery. The lung or heart was exposed by means of a thoracotomy. The pericardium was removed using fine forceps for the heart implantation. The adhesive implant (10 mm in width and 10 mm in length) was applied to the lung ($n$ = 4) or heart ($n$ = 4) surface by gently pressing with a surgical spatula or fingertip. The non-adhesive implant (10 mm in width and 10 mm in length) was implanted to the lung ($n$ = 4) or heart ($n$ = 4) surface

by sutures at the corners of the samples (8-0 Prolene, Ethicon)[24]. The muscle and skin incisions were closed with sutures (4-0 Vicryl, Ethicon). The animal was ventilated with 100% oxygen until normal breathing resumed. On days 28 and 84 post-implantation, the animals were euthanized by $CO_2$ inhalation. Lung or heart tissues of interest were excised and fixed in 10% formalin for 24 h for histological and immunofluorescence analysis. All animals in the study survived and were kept in normal health conditions on the basis of daily monitoring by the MIT DCM veterinarian staff.

### In vivo intraperitoneal implantation in mouse model
All animal studies on mice were approved by the MIT Committee on Animal Care, and all surgical procedures and postoperative care were supervised by the MIT DCM veterinary staff. The mice housing room temperature was set at 21 °C with the room monitoring alarms set at ±2 °C, and relative humidity was maintained at 30–70% with a 12 h light/12 h dark cycle.

Immunocompetent C57BL/6 mice (female and male, 18–25 g, 6–8 weeks, Jackson Laboratory) or humanized HuCD34-NCG mice (female, 18–25 g, 16–18 weeks, Charles River Laboratories) were anaesthetized with 2–3% isoflurane, and then the abdomen was shaved and cleaned using betadine and 70% ethanol. A 1-cm incision was made along the abdomen midline and the abdominal wall was exposed by means of a laparotomy. The adhesive implant (5 mm in width and 5 mm in length) or non-adhesive implant (5 mm in width and 5 mm in length) was applied to the abdominal wall ($n$ = 6 per group for C57BL/6 mice; $n$ = 5 per group for HuCD34-NCG mice) by gently pressing. Both PVA-based and chitosan-based samples were used for C57BL/6 mice. Only PVA-based samples were used for HuCD34-NCG mice. The abdominal wall muscle and skin incisions were closed with sutures (5-0 Vicryl, Ethicon). On days 14 and 28 post-implantation, the abdominal wall of interest was excised and fixed in 10% formalin overnight for histological analysis.

### In vivo intraperitoneal implantation in porcine model
All animal studies on pigs were approved by the Mayo Clinic institutional animal care and use committee at Rochester.

The female domestic pigs (female, 50 kg, 20 weeks, Manthei Hog Farm) were placed in dorsal recumbency, and the abdominal region was clipped and prepared aseptically. A blade was used to incise the ventral midline and extended using electrocautery when necessary. The linea alba was incised, and the peritoneum was bluntly entered, with the incision extended to match the skin incision. The small intestine was exteriorized and moist lap sponges were used for isolation. Then, the adhesive implant or non-adhesive implant was applied and adhered to the surface of the abdominal wall and small intestine ($n$ = 4 for each group). The small intestine was thoroughly lavaged and returned to the abdomen. Then, the entire abdominal cavity was lavaged and suctioned, and the celiotomy incision was closed. On day 7 post-implantation, the animals were humanely euthanized, and the abdominal wall and small intestine of interest were excised and fixed in 10% formalin for 24 h for histological analyses. All animals in the study survived and were kept in normal health conditions on the basis of daily monitoring by the Mayo Clinic Rochester veterinarian staff.

### In vivo electrophysiological study
Before implantation, the adhesive and non-adhesive implantable electrodes were prepared using aseptic techniques and were further disinfected for 1 h under UV. For in vivo epicardial electrode implantation, the animals were anaesthetized using isoflurane (2 to 3% isoflurane in oxygen) in an anaesthetizing chamber before the surgery, and anaesthesia was maintained using a nose cone throughout the surgery. Chest and back hair were removed, and endotracheal intubation was carried out, connecting the animals to a mechanical ventilator (RoVent, Kent Scientific). The animals were placed on a heating pad for the duration of the surgery. The heart was exposed by means of a thoracotomy and

the pericardium was removed using fine forceps for the epicardial implantation. The adhesive implantable electrodes were applied to the left ventricular surface ($n$ = 6) by gently pressing with a surgical spatula or fingertip. The non-adhesive implantable electrodes were implanted to the left ventricular surface ($n$ = 6) by sutures at the corners of the samples (8-0 Prolene, Ethicon). The lead wire was then tunnelled subcutaneously from a ventral exit site close to the left fourth intercostal space to the dorsal side. The dorsal end of the lead wire was inserted through a subcutaneous port. The subcutaneous port was placed by interrupted sutures (4-0 Vicryl, Ethicon) between the shoulder blades of the animal and covered by a protective aluminium cap (VABRC, Instech Laboratories). The muscle and skin incisions were closed with sutures (4-0 Vicryl, Ethicon). The animal was ventilated with 100% oxygen until autonomous breathing was regained.

On days 0, 3, 7, 14, 28, 56 and 84 post-implantation, each animal was anaesthetized and connected to the data acquisition hardware (PowerLab, AD Instrument) and software (LabChart Pro 7, AD Instrument) for electrophysiological recording and stimulation by the implanted electrodes. For electrophysiological recording, the data acquisition hardware was connected to the implanted electrodes through the dorsal subcutaneous port. Epicardial signals were recorded to evaluate the R-wave amplitude. For electrophysiological stimulation, an external stimulator (FE180, AD Instrument) was connected to the implanted electrodes through the dorsal subcutaneous port. Unipolar rectangular current pulses (0.5 ms, 0–3 mA, 5–7 Hz) were used for continuous ventricular pacing and the surface electrocardiogram was monitored to evaluate the capture threshold at the same time. On days 28 and 84 post-implantation, the animals were euthanized by $CO_2$ inhalation. Heart tissues of interest were excised and fixed in 10% formalin for 24 h for histological analysis. All animals in the study survived and were kept in normal health conditions on the basis of daily monitoring by the MIT DCM veterinarian staff.

### Immunofluorescence analysis
The expression of targeted markers (αSMA, CD68, CD3, CD206, iNOS, vimentin, neutrophil elastase) was analysed after the immunofluorescence staining of the collected tissues. Before the immunofluorescence analysis, the paraffin-embedded fixed tissues were sliced and prepared into slides. The slides were deparaffinized and rehydrated with deionized water. Antigen retrieval was carried out using the steam method during which the slides were steamed in IHC-Tek Epitope Retrieval Solution (IW-1100) for 35 min and then cooled for 20 min. Then the slides were washed in three changes of PBS for 5 min per cycle. After washing, the slides were incubated in primary antibodies (1:200 mouse anti-αSMA (ab7817, Abcam); 1:200 mouse anti-CD68 (ab201340, Abcam); 1:100 rabbit anti-CD3 (ab5690, Abcam); 1:1,000 rabbit anti-CD206 (ab64693, Abcam); 1:500 mouse anti-vimentin (ab8978, Abcam); 1:2,000 rabbit anti-iNOS (ab283655, Abcam); 1:200 mouse anti-iNOS (GTX60599, GeneTex); 1:50 rabbit anti-neutrophil elastase (bs-6982R, Bioss)) diluted with IHC-Tek antibody diluent for 1 h at room temperature. The slides were then washed three times in PBS and incubated with Alexa Fluor 488-labelled anti-rabbit or anti-mouse secondary antibody (1:200, Jackson Immunoresearch) or Alexa Fluor 594-labelled donkey anti-mouse secondary antibody (1:200, Jackson Immunoresearch) for 30 min. The slides were washed in PBS and then counterstained with propidium iodide solution for 20 min. A laser confocal microscope (SP8, Leica) was used for image acquisition. ImageJ (version 2.1.0) was used to quantify the number of cells in the collagenous layer at the implant–tissue interface from the immunofluorescence images[34] (500 μm width of the field of view). All analyses were blinded with respect to the experimental conditions.

### Luminex quantification analysis
On days 3 and 7 post-implantation, the abdominal muscle wall of interest was collected. The collected samples were snap-frozen in liquid

nitrogen and homogenized on a TissueLyser LT (Qiagen) following the manufacturer's instructions. A Luminex multiplex assay was used to measure the concentrations of immune-response-related cytokines and chemokines (RECYTMAG-65K, Milliplex). Values per sample were normalized to the total protein content and expressed as picograms per total milligram of protein (Supplementary Table 1).

## qPCR analysis

RNA was isolated from the samples snap-frozen in liquid nitrogen immediately after excision using the TRIzol protocol (Invitrogen). All samples were homogenized and normalized by loading 1 μg of total RNA in all cases for reverse transcription using a Super-Script First Strand cDNA Synthesis Kit (Invitrogen). Complementary DNA (1:20 dilution) was amplified by qPCR with the following primers: *Mrc1* (5′-AACTTCATCTGCCAGCGACA-3′; reverse: 5′-CGT GCCTCTTTCCAGGTCTT-3′), *Tgfb1* (5′-AGTGGCTGAACCAAGGAGAC-3′; reverse: 5′-CCTCGACGTTTGGGACTGAT-3′), *Nos2* (5′-TGGTGAGGG GACTGGACTTT-3′; reverse: 5′-CCAACTCTGCTGTTCTCCGT-3′), *Cd86* (5′-AGACATGTGTAACCTGCACCAT-3′; reverse: 5′-TACGAGC TCACTCGGGCTTA-3′), *S100a8* (5′-CGAAGAGTTCCTTGTGTTGGTG-3′; reverse: 5′-AGCTCTGTTACTCCTTGTGGC-3′), *Ly6c* (5′-ACCTG GTCACAGAGAGGAAGT-3′; reverse: 5′-AGCAGTTAGCATTAAG TGGGACT-3′), *Il10* (5′-TTGAACCACCCGGCATCTAC-3′; reverse: 5′-CCAAGGAGTTGCTCCCGTTA-3′), *Cd11b* (5′-GACTCCGCATT TGCCCTACT-3′; reverse: 5′-GCTGCCCACAATGAGTGGTA-3′) and glyceraldehyde-3-phosphate dehydrogenase (*Gapdh*) (5′-CAC CATCTTCCAGGAGCGAG-3′; reverse: 5′-CCACGACATACTCAGCACCA-3′). Samples were incubated for 10 min at 95 °C for 15 s and at 60 °C for 1 min in the real-time cycler Agilent MX3000P. *Gapdh* was used as the reference gene for normalization and analysis. The comparative CT (ΔΔCT) method was used for relative quantification of gene expression.

## RNA-sequencing analysis

RNA extraction, library preparation and sequencing reactions were conducted at GENEWIZ. Total RNA was extracted using the Qiagen RNeasy Plus Universal mini kit following the manufacturer's instructions (Qiagen). Extracted RNA samples were quantified using the Qubit 2.0 Fluorometer (Life Technologies) and RNA integrity was checked on Agilent TapeStation 4200 (Agilent Technologies). RNA-sequencing libraries were prepared using the NEBNext Ultra RNA Library Prep Kit for Illumina following the manufacturer's instructions (NEB). Briefly, mRNAs were first enriched with Oligo(dT) beads. Enriched mRNAs were fragmented for 15 min at 94 °C. First-strand and second-strand cDNAs were subsequently synthesized. cDNA fragments were end-repaired and adenylated at the 3′ ends, and universal adaptors were ligated to cDNA fragments, followed by index addition and library enrichment by limited-cycle PCR. The sequencing libraries were validated on the Agilent TapeStation (Agilent Technologies), and quantified using the Qubit 2.0 Fluorometer (Invitrogen) as well as by qPCR (KAPA Biosystems). The sequencing libraries were clustered on one lane of a flow cell. After clustering, the flow cell was loaded on the Illumina HiSeq 4000 instrument and the samples were sequenced using a 2 × 150-base-pair paired end configuration. Image analysis and base calling were conducted by the HiSeq Control Software. Raw sequence data (.bcl files) generated from Illumina HiSeq were converted into fastq files and de-multiplexed using Illumina's bcl2fastq 2.17 software. One mismatch was allowed for index sequence identification.

Read quality was evaluated using FastQC, and data were pre-processed with Cutadapt[35] for adaptor removal following best practices[36]. Gene expression against the mRatBN7.2 transcriptome (Ensembl release 104)[37] was quantified with STAR[38] and featureCounts[39]. Differential gene expression analysis was carried out using DESeq2 (ref. 40), and

ClusterProfiler[41] was used for functional enrichment investigations. Genes with $\log_2$[fold change] ≥ 1 and false discovery rate ≤ 0.05 were considered statistically significant.

## Statistical analysis

GraphPad Prism (version 9.2.0) was used to assess the statistical significance of all comparison studies in this work. Data distribution was assumed to be normal for all parametric tests, but not formally tested. In the statistical analysis for comparison between multiple groups, one-way analysis of variance followed by Bonferroni's multiple comparison test was conducted with the significance thresholds at *$P < 0.05$, **$P ≤ 0.01$, ***$P ≤ 0.001$ and ****$P < 0.0001$. In the statistical analysis of two groups, the two-sided unpaired $t$-test was used with the significance thresholds at *$P < 0.05$, **$P ≤ 0.01$, ***$P ≤ 0.001$ and ****$P < 0.0001$.

## Reporting summary

Further information on research design is available in the Nature Portfolio Reporting Summary linked to this article.

## Data availability

All data supporting the findings of this study are available within the article and its Supplementary Information. The RNA-sequencing data generated in the present study were deposited in the Gene Expression Omnibus with accession number GSE198219. Additional raw data generated in this study are available from the corresponding authors upon reasonable request. Source data are provided with this paper.

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

**Acknowledgements** We thank the Koch Institute Swanson Biotechnology Center for technical support, specifically the Hope Babette Tang (1983) Histology Core for histological processing and the Peterson (1957) Nanotechnology Materials Core for transmission electron microscopy imaging, and Z. Wei and Q. Zhou for discussions. This work is supported by the National Institute of Health (1-R01HL167947-01 and 1-R01-HL153857-01), Department of Defense Congressionally Directed Medical Research Programs (PR200524P1) and the National Science Foundation (EFMA-1935291).

**Author contributions** X.Z. proposed the idea. H.Y. and J.W. developed the adhesive interface. J.W., H.Y. and X.Z. designed the study. J.W. and H.Y. carried out the in vitro and in vivo rat studies. J.W. carried out the in vivo mice studies. T.L.S. and L.G.G. carried out the in vivo porcine study. J.D., J.W. and H.Y. carried out the in vivo electrophysiological studies. J.W. carried out the immunofluorescence and qPCR analyses. G.T. and A.V. carried out the cytokine and sequencing analyses. R.T.B. evaluated the histological data. J.C. provided support for immunological analyses and edited the manuscript. H.Y. and J.W. prepared figures with input from all authors. J.W., H.Y. and X.Z. prepared the manuscript and all authors reviewed and edited the manuscript. H.Y. and X.Z. supervised the study.

**Competing interests** H.Y. and X.Z. have a financial interest in SanaHeal. X.Z. has a financial interest in SonoLogi. J.W., J.D., H.Y. and X.Z. are inventors of a patent application that covers the adhesive anti-fibrotic interfaces. The other authors declare no competing interests.

**Additional information**
**Correspondence and requests for materials** should be addressed to Hyunwoo Yuk or Xuanhe Zhao.

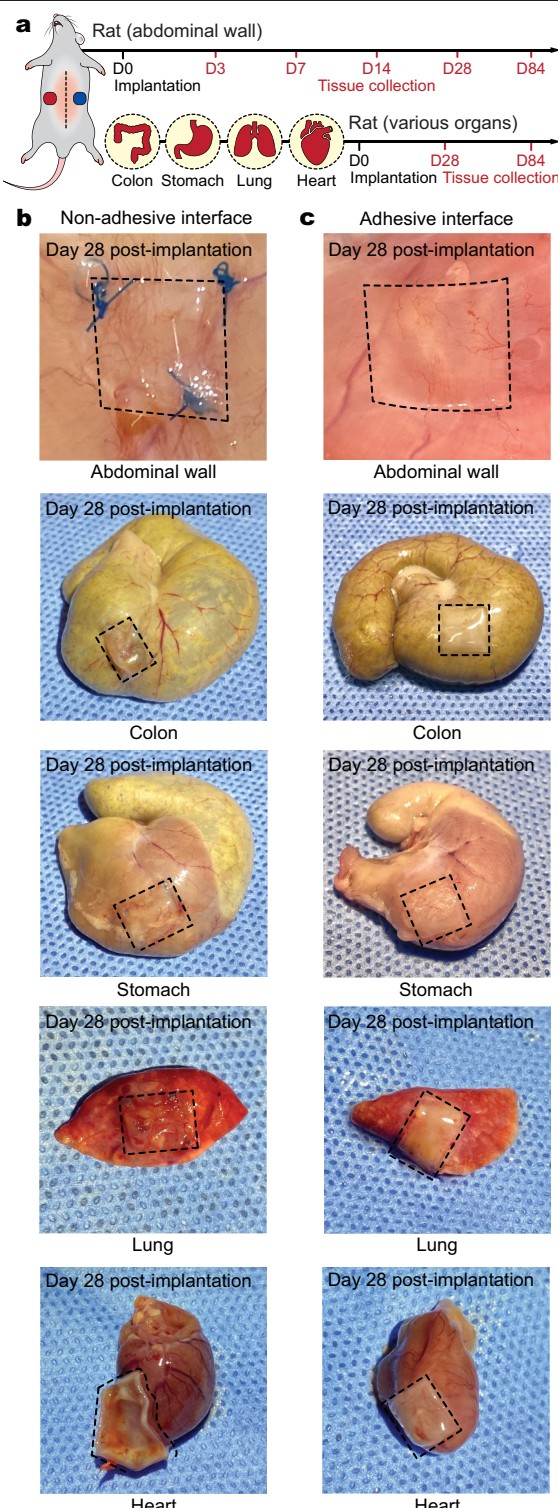

**Extended Data Fig. 1 | In vivo implantation of the adhesive and non-adhesive implants to various organs. a**, Schematic illustrations for the in vivo rat studies. **b,c**, Photographs of various organs collected on day 28 post-implantation for the non-adhesive implant (**b**) and the adhesive implant (**c**). Black dotted lines in photographs indicate the boundary of implants.

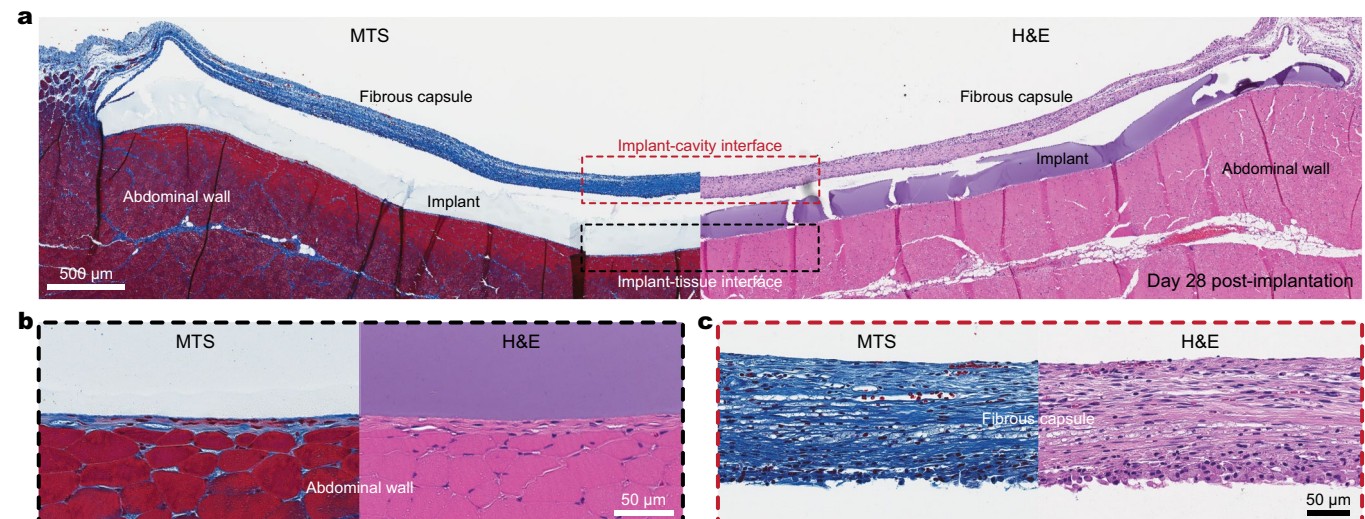

**Extended Data Fig. 2 | Adhesive implant histology. a**, Representative histology images stained with Masson's trichrome (MTS, left) and haematoxylin and eosin (H&E, right) of the adhesive implant collected on day 28 post-implantation to the abdominal wall. Black and red dotted areas indicate the implant-tissue interface and the implant-abdominal cavity interface, respectively. **b,c**, Representative histology images stained with MTS (left) and H&E (right) of the implant-tissue interface (**b**) and implant-cavity interface (**c**) for the adhesive implant collected on day 28 post-implantation to the abdominal wall. The experiment was repeated independently (*n* = 4) with similar results.

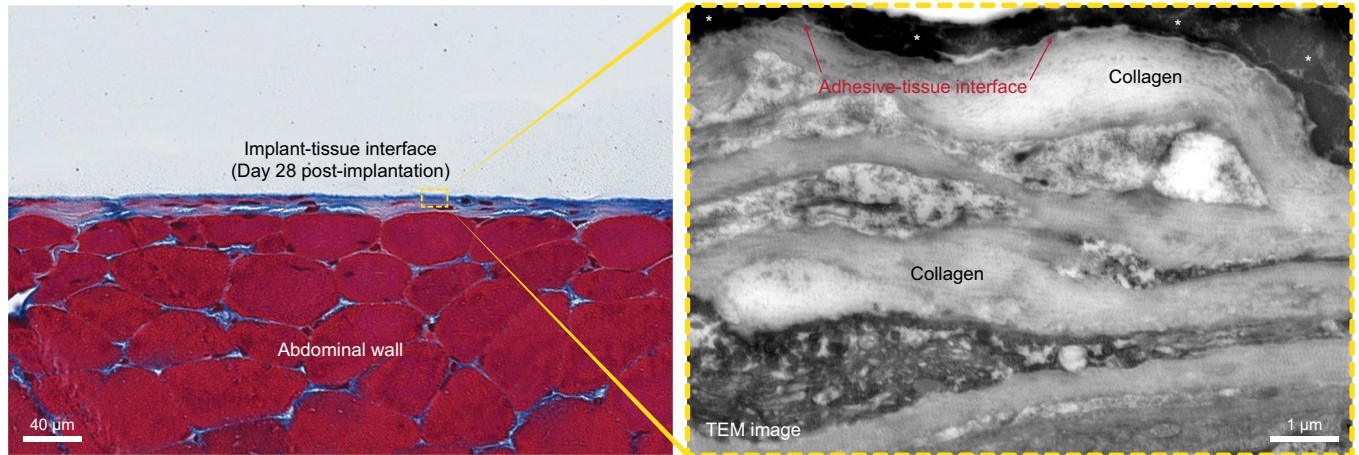

**Extended Data Fig. 3 | TEM image of the adhesive implant-tissue interface.**
Representative histology image stained with Masson's trichrome (left) and TEM image (right) of the adhesive implant collected on day 28 post-implantation to the abdominal wall. *In images indicates the implant. The experiment was repeated independently ($n = 4$) with similar results.

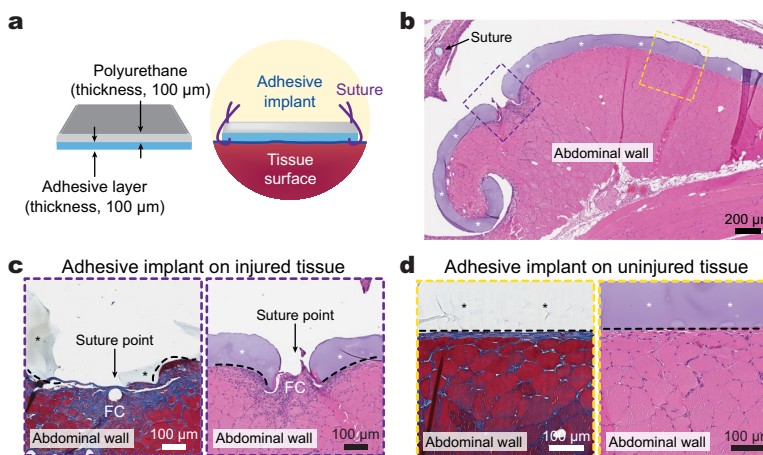

**Extended Data Fig. 4 | Adhesive implant-tissue interface with sutures.**
**a**, Schematic illustrations of the adhesive implant with sutures at the corners.
**b**, Representative histology image stained with haematoxylin and eosin (H&E) for the adhesive implant with sutures on the abdominal wall collected on day 28 post-implantation. **c,d**, Representative histology images stained with Masson's trichrome (MTS, left) and H&E (right) for the suture point (**c**) and the intact adhesive-tissue interface (**d**) collected on day 28 post-implantation to the abdominal wall. *In images indicates the implant; black dotted lines indicate the implant-tissue interface. FC, fibrous capsule. The experiment in **b**–**d** was repeated independently (*n* = 6) with similar results.

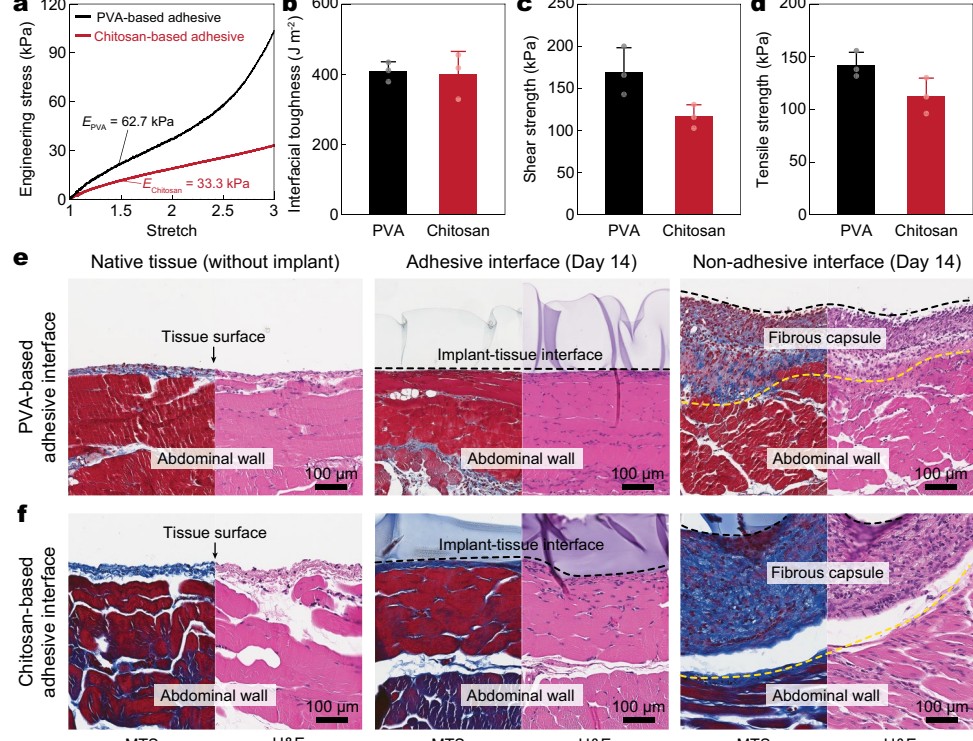

**Extended Data Fig. 5 | Chitosan-based adhesive interface. a**, Engineering stress versus stretch curves for the PVA-based and chitosan-based adhesive interfaces. $E_{PVA}$, Young's modulus of the PVA-based adhesive interface; $E_{chitosan}$, Young's modulus of the chitosan-based adhesive interface. **b–d**, Interfacial toughness (**b**), shear strength (**c**), and tensile strength (**d**) of the PVA-based and chitosan-based adhesive interfaces on ex vivo porcine skin. **e,f**, Representative histology images stained with Masson's trichrome (MTS) and haematoxylin and eosin (H&E) for native tissue (left), adhesive implant (middle), and non-adhesive implant (right) collected on day 14 post-implantation to the abdominal wall based on the PVA-based adhesive interface (**e**) and the chitosan-based adhesive interface (**f**). Black and yellow dotted lines in the images indicate the implant-tissue interface and the fibrous capsule-tissue interface, respectively. Values in **b–d** represent the mean and the standard deviation ($n = 3$, independent samples). The experiment in **e,f** was repeated independently ($n = 4$ per group) with similar results.

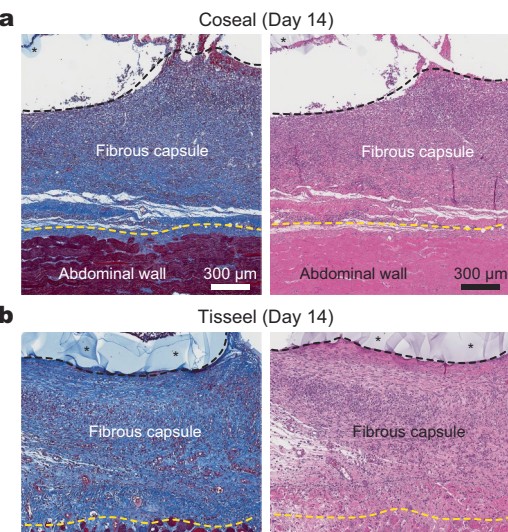

**Extended Data Fig. 6 | Adhesive interface by commercially-available tissue adhesives. a,b,** Representative histology images stained with Masson's trichrome (left) and haematoxylin and eosin (right) for the implant integrated to the abdominal wall surface by Coseal (**a**) and Tisseel (**b**) collected on day 14 post-implantation. *In images indicates the implant; black dotted lines indicate the implant-tissue interface; yellow dotted lines indicate the fibrous capsule-tissue interface. The experiment was repeated independently (*n* = 6 per group) with similar results.

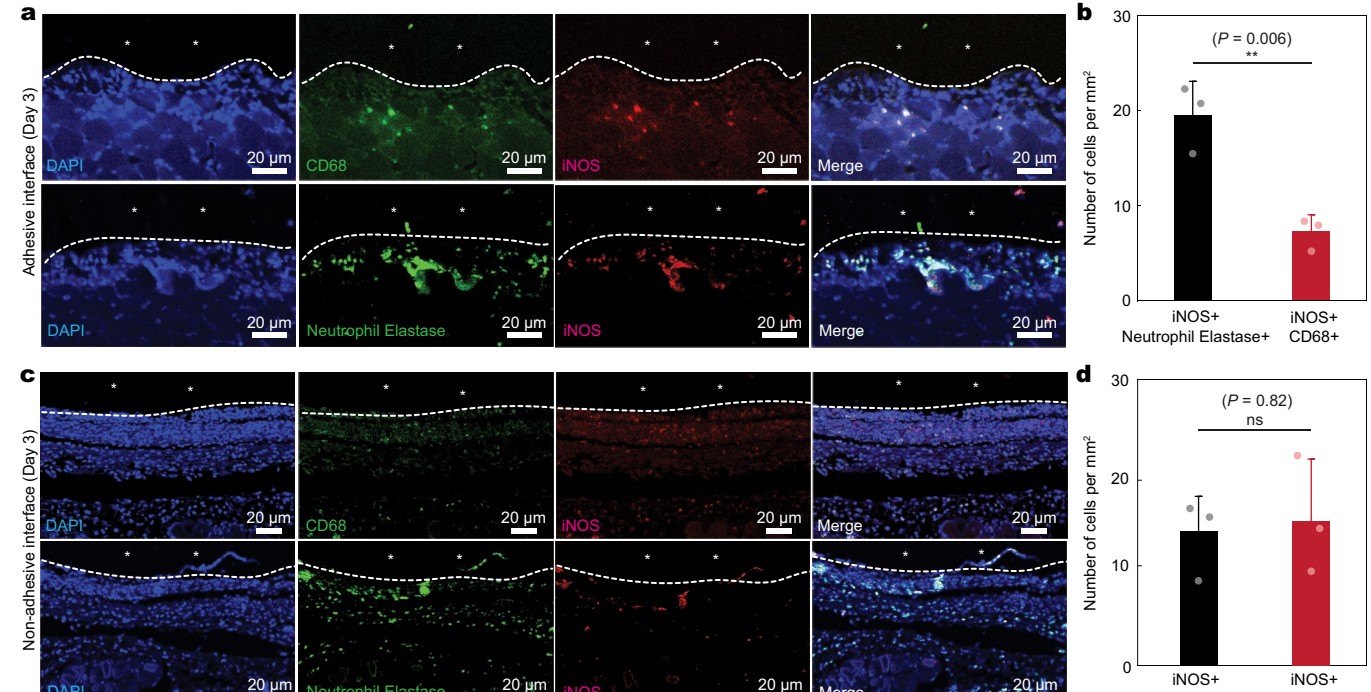

**Extended Data Fig. 7 | Immunofluorescence analysis of iNOS+ cells at the implant-tissue interface. a**, Representative immunofluorescence images at the adhesive implant-tissue interface on day 3 post-implantation to the abdominal wall. **b**, Quantification of iNOS + /neutrophil elastase+ and iNOS + /CD68+ cells per unit area on day 3 post-implantation for the adhesive implant-tissue interface. **c**, Representative immunofluorescence images at the non-adhesive implant-tissue interface on day 3 post-implantation to the abdominal wall. **d**, Quantification of iNOS + /neutrophil elastase+ and iNOS + /CD68+ cells per unit area on day 3 post-implantation for the non-adhesive implant-tissue

interface. In immunofluorescence images, cell nuclei are stained with 4′,6-diamidino-2-phenylindole (DAPI, blue); green fluorescence corresponds to the expression of macrophage (CD68) and neutrophil (neutrophil elastase); red fluorescence corresponds to the expression of iNOS. *In images indicates the implant; white dotted lines in images indicate the implant-tissue interface. Values in **b,d** represent the mean and the standard deviation (*n* = 3 implants; independent biological replicates). Statistical significance and *P* values are determined by two-sided unpaired *t*-tests; ns, not significant; **$P \leq 0.01$.

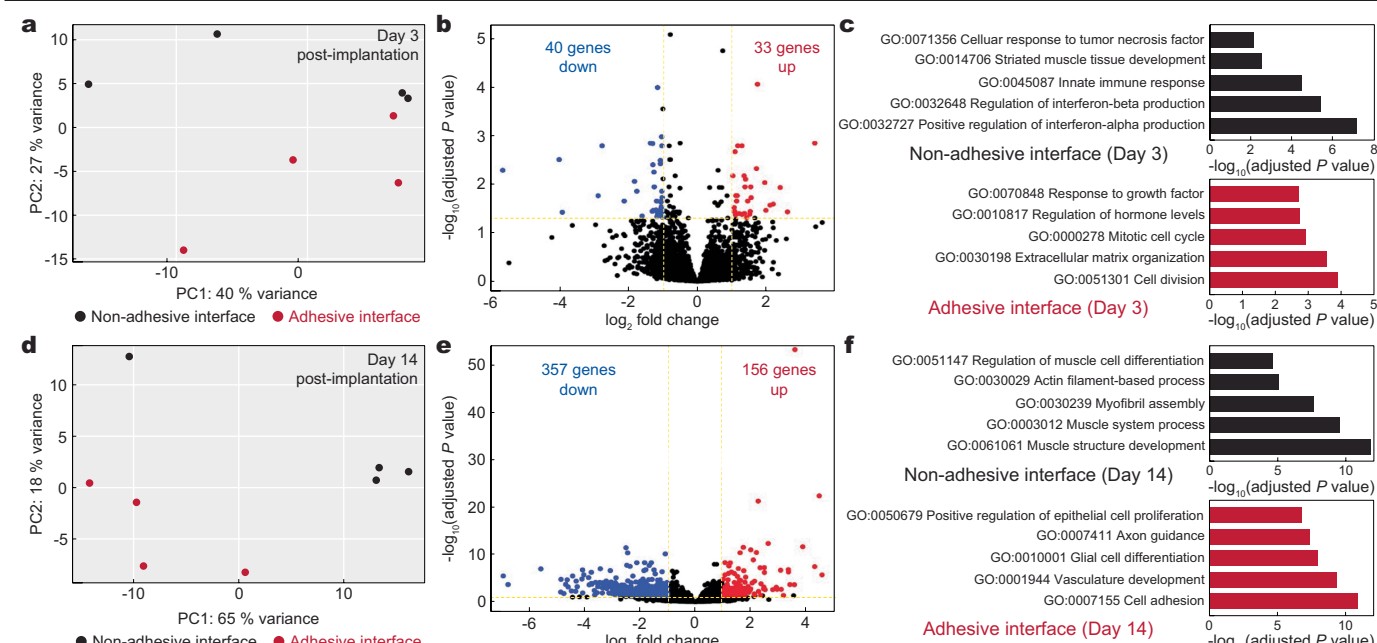

**Extended Data Fig. 8 | Transcriptomic analysis of adhesive and non-adhesive implant-tissue interfaces. a**, Principal component analysis (PCA) plot illustrating the variances of the adhesive (red dots, $n = 4$) and non-adhesive (black dots, $n = 4$) implant-tissue interface dataset collected on day 3 post-implantation to the abdominal wall. **b**, Volcano plot displaying the gene expression profiles for the non-adhesive and adhesive implant-tissue interfaces collected on day 3 post-implantation to the abdominal wall. Coloured (blue and red) data points represent genes that meet the threshold of fold change (FC) above 1 or under −1, false discovery rate (FDR) < 0.05. Blue and red coloured dots indicate down- and up-regulated genes in the adhesive implant-tissue interface compared to the non-adhesive implant-tissue interface, respectively. **c**, Top five enriched processes from Gene Ontology (GO) enrichment analysis of differentially expressed genes in the non-adhesive (black) and adhesive (red) implant-tissue interfaces collected on day 3 post-

implantation to the abdominal wall. **d**, PCA plot illustrating the variances of the adhesive (red dots, $n = 4$) and non-adhesive (black dots, $n = 4$) implant-tissue interface dataset collected on day 14 post-implantation to the abdominal wall. **e**, Volcano plot displaying the gene expression profiles for the non-adhesive and adhesive implant-tissue interfaces collected on day 14 post-implantation to the abdominal wall. Coloured (blue and red) data points represent genes that meet the threshold of fold change (FC) above 1 or under −1, false discovery rate (FDR) < 0.05. Blue and red coloured dots indicate down- and up-regulated genes in the adhesive implant-tissue interface compared to the non-adhesive implant-tissue interface, respectively. **f**, Top five enriched processes from Gene Ontology (GO) enrichment analysis of differentially expressed genes in the non-adhesive (black) and adhesive (red) implant-tissue interfaces collected on day 14 post-implantation to the abdominal wall. The $P$ values were determined by one-sided Fisher's exact test and adjusted by Storey's correction method.

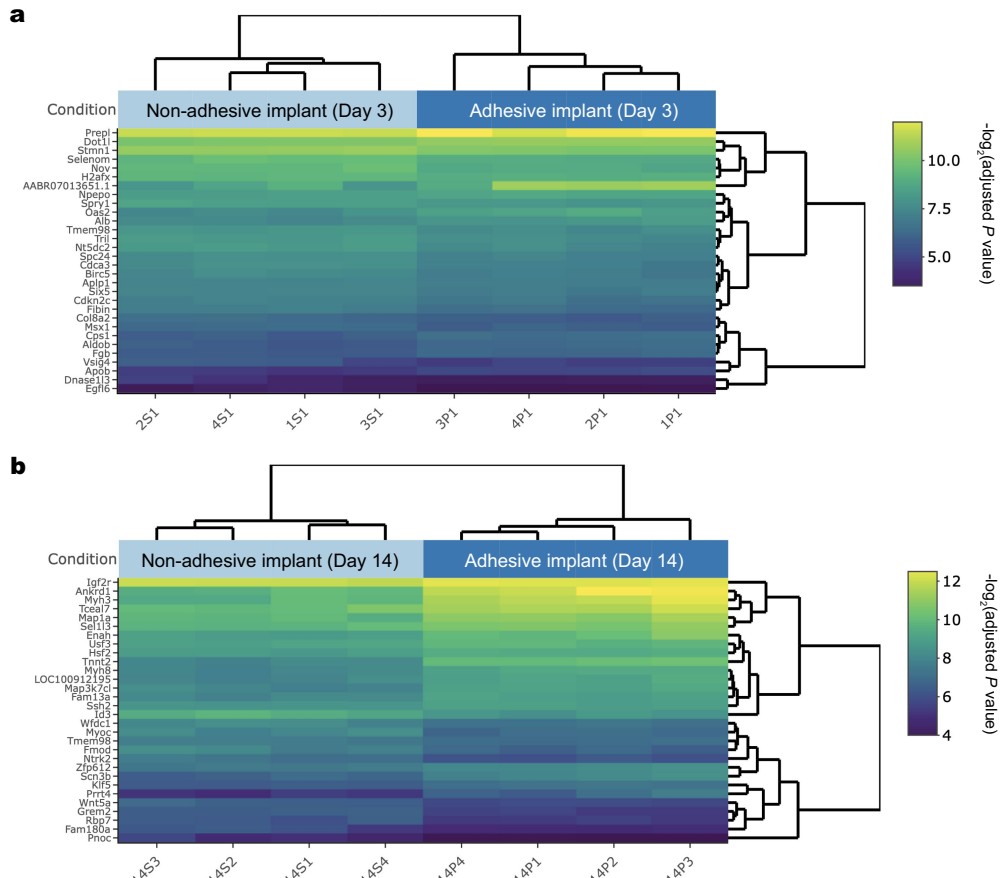

**Extended Data Fig. 9 | Visualization of RNA sequencing results.**
**a**,**b**, Bi-clustering heatmap to visualize the expression profiles of the top 30 differentially expressed genes sorted by their adjusted *P* value by plotting their log2 transformed expression values in samples day 3 (**a**) and day 14 (**b**) post-implantation. Dendrograms were drawn from Ward hierarchical clustering. The *P* values were determined by one-sided Fisher's exact test and adjusted by Storey's correction method.

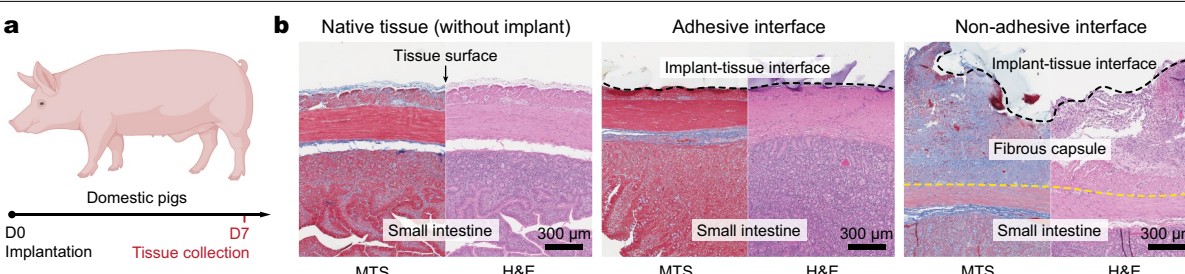

**Extended Data Fig. 10 | Adhesive anti-fibrotic interfaces in porcine model. a**, Schematic illustration for the study design based on the porcine model. **b**, Representative histology images stained with Masson's trichrome (MTS) and haematoxylin and eosin (H&E) for native tissue (left), adhesive implant (middle), and non-adhesive implant (right) collected on 7 days post-implantation to the small intestine. Black dotted lines in images indicate the implant-tissue interface; yellow dotted lines in images indicate the fibrous capsule-tissue interface. The experiment was repeated independently (*n* = 4 per group) with similar results. The graphic of the pig in **a** was created with BioRender.com.

# nature research

| | |
|---|---|

# Reporting Summary

Nature Research wishes to improve the reproducibility of the work that we publish. This form provides structure for consistency and transparency in reporting. For further information on Nature Research policies, see Authors & Referees and the Editorial Policy Checklist.

## Statistics

For all statistical analyses, confirm that the following items are present in the figure legend, table legend, main text, or Methods section.

| n/a | Confirmed | |
|---|---|---|
| ☐ | ☒ | The exact sample size (*n*) for each experimental group/condition, given as a discrete number and unit of measurement |
| ☐ | ☒ | A statement on whether measurements were taken from distinct samples or whether the same sample was measured repeatedly |
| ☐ | ☒ | The statistical test(s) used AND whether they are one- or two-sided<br>*Only common tests should be described solely by name; describe more complex techniques in the Methods section.* |
| ☒ | ☐ | A description of all covariates tested |
| ☐ | ☒ | A description of any assumptions or corrections, such as tests of normality and adjustment for multiple comparisons |
| ☐ | ☒ | A full description of the statistical parameters including central tendency (e.g. means) or other basic estimates (e.g. regression coefficient) AND variation (e.g. standard deviation) or associated estimates of uncertainty (e.g. confidence intervals) |
| ☐ | ☒ | For null hypothesis testing, the test statistic (e.g. *F*, *t*, *r*) with confidence intervals, effect sizes, degrees of freedom and *P* value noted<br>*Give P values as exact values whenever suitable.* |
| ☒ | ☐ | For Bayesian analysis, information on the choice of priors and Markov chain Monte Carlo settings |
| ☒ | ☐ | For hierarchical and complex designs, identification of the appropriate level for tests and full reporting of outcomes |
| ☒ | ☐ | Estimates of effect sizes (e.g. Cohen's *d*, Pearson's *r*), indicating how they were calculated |

*Our web collection on statistics for biologists contains articles on many of the points above.*

## Software and code

Policy information about availability of computer code

| Data collection | No software used for data collection. |
|---|---|
| Data analysis | Microscopic images were analyzed by using ImageJ (Version: 2.1.0). All statistical analyses were performed by using GraphPad Prism (Version: 9.2.0). Raw sequencing data (.bcl files) generated from Illumina HiSeq were converted into fastq files and de-multiplexed using Illumina's bcl2fastq software (version 2.17). |

For manuscripts utilizing custom algorithms or software that are central to the research but not yet described in published literature, software must be made available to editors/reviewers. We strongly encourage code deposition in a community repository (e.g. GitHub). See the Nature Research guidelines for submitting code & software for further information.

## Data

Policy information about availability of data

All manuscripts must include a data availability statement. This statement should provide the following information, where applicable:
- Accession codes, unique identifiers, or web links for publicly available datasets
- A list of figures that have associated raw data
- A description of any restrictions on data availability

All data supporting the findings of this study are available within the Article and its Supplementary Information. The RNA-seq data generated in the present study were deposited in the Gene Expression Omnibus (GEO) with accession number 'GSE198219'. Additional raw data generated in this study are available from the corresponding authors upon reasonable request.

# Field-specific reporting

Please select the one below that is the best fit for your research. If you are not sure, read the appropriate sections before making your selection.

☒ Life sciences  ☐ Behavioural & social sciences  ☐ Ecological, evolutionary & environmental sciences

For a reference copy of the document with all sections, see nature.com/documents/nr-reporting-summary-flat.pdf

# Life sciences study design

All studies must disclose on these points even when the disclosure is negative.

| | |
|---|---|
| Sample size | The sample size for rodent and porcine studies were determined based on the literature with similar studies (Wu et al., Science Translational Medicine 14, eabh2857 (2022)). |
| | For all rats in vivo studies, the appropriate sample size (n = 4 per each time point & group) were conducted to investigate foreign body reactions with implant on various organs used. |
| | In vivo experiments on HuCD34-NCG mice (n = 5) and C57BL/6 mice (n = 6 per each time point) were conducted to investigate foreign body reactions with adhesive or non adhesive implant. |
| | In vivo experiments on pig (n = 4) were conducted to investigate foreign body reactions with implant. |
| Data exclusions | No animal was excluded. |
| Replication | In vivo studies were reliably reproduced based on comparable histological assessment for each case by the blinded pathologist. All in vivo studies were independently performed with at least 1 day between surgeries. All attempts at replication were successful. |
| Randomization | All the tests were performed with randomly allocated experimental groups. |
| Blinding | All histological assessments were conducted by the blinded pathologist based on randomly mixed histological slides without informing type or study group of samples. All other measurements were conducted in a blinded fashion. |

# Reporting for specific materials, systems and methods

We require information from authors about some types of materials, experimental systems and methods used in many studies. Here, indicate whether each material, system or method listed is relevant to your study. If you are not sure if a list item applies to your research, read the appropriate section before selecting a response.

## Materials & experimental systems

| n/a | Involved in the study |
|---|---|
| ☐ | ☒ Antibodies |
| ☒ | ☐ Eukaryotic cell lines |
| ☒ | ☐ Palaeontology |
| ☐ | ☒ Animals and other organisms |
| ☒ | ☐ Human research participants |
| ☒ | ☐ Clinical data |

## Methods

| n/a | Involved in the study |
|---|---|
| ☒ | ☐ ChIP-seq |
| ☒ | ☐ Flow cytometry |
| ☒ | ☐ MRI-based neuroimaging |

# Antibodies

| | |
|---|---|
| Antibodies used | Primary antibodies: Mouse anti-αSMA (ab7817, Abcam); Mouse anti-CD68 (ab201340, Abcam); Rabbit anti-CD3 (ab5690, Abcam); Rabbit anti-CD206 (ab64693, Abcam); Mouse anti-vimentin (ab8978, Abcam); Rabbit anti-iNOS (ab283655, Abcam); Mouse anti-iNOS (GTX60599, GeneTex); Rabbit anti-neutrophil elastase (bs-6982R, Bioss). |
| | Secondary antibodies: Alexa Fluor 488 labeled anti-rabbit or anti-mouse secondary antibody (315-545-003, Jackson Immunoresearch), or Alexa Fluor 594 labeled donkey anti-mouse secondary antibody (715-586-151, Jackson Immunoresearch). |
| Validation | All antibodies are commercially available and have been tested by the manufacturer. Vendors and catalog numbers are listed above and validation can be found there. |
| | Mouse anti-aSMA (ab7817, Abcam): This monoclonal antibody recognizes aSMA. Manufacturer-validated to react with Mouse, Rat, Rabbit, Human, Pig aSMA (https://www.abcam.com/alpha-smooth-muscle-actin-antibody-1a4-ab7817.html). |
| | Mouse anti-CD68 (ab201340, Abcam): This monoclonal antibody recognizes CD68. Manufacturer-validated to react with Mouse, Rat, Human CD68 (https://www.abcam.com/cd68-antibody-c68684-ab201340.html). |

Rabbit anti-CD3 (ab5690, Abcam): This polyclonal antibody recognizes to CD3. Manufacturer-validated to react with Mouse, Rat, Human CD3 (https://www.abcam.com/cd3-antibody-ab5690.html).

Rabbit anti-iNOS (ab283655, Abcam): This multiclonal antibody recognizes to iNOS. Manufacturer-validated to react with Mouse, Rat, Human iNOS (https://www.abcam.com/inos-antibody-rm1017-ab283655.html)

Rabbit anti-CD206 (ab64693, Abcam): This polyclonal antibody recognized to CD206. Manufacturer-validated to react with Mouse, Rat, Human CD206 (https://www.abcam.com/mannose-receptor-antibody-ab64693.html).

Mouse anti-Vimentin (ab8978, Abcam): This monoclonal antibody recognized to Vimentin. Manufacturer-validated to react with Mouse, Rat, Human Vimentin (https://www.abcam.com/vimentin-antibody-rv202-cytoskeleton-marker-ab8978.html).

Mouse anti-iNOS (GTX60599, GeneTex): This monoclonal antibody recognized to iNOS. Manufacturer-validated to react with Mouse, Rat, Human iNOS (https://www.genetex.com/Product/Detail/iNOS-antibody-4E5/GTX60599?utm_source=listng&utm_medium=click&utm_id=AntibodyResource)

Rabbit anti-Neutrophil Elastase (bs-6982R-A488, BiossAntibodies): This polyclonal antibody recognized to neutrophil elastase. Manufacturer-validated to react with Human, Mouse, Rat neutrophil elastase (https://www.biossusa.com/products/bs-6982r-a488)

# Animals and other organisms

Policy information about studies involving animals; ARRIVE guidelines recommended for reporting animal research

| Laboratory animals | Female and male Sprague Dawley rats (12 weeks, 225-275g weight), were purchased from Charles River Laboratories.<br>Female and male C57BL/6 mice (6-8 weeks, 18-25g) were purchased from Jackson Laboratory.<br>Female HuCD34-NCG mice (16-18 weeks, 18-25g) were purchased from Charles River Laboratories.<br>Female Domestic pigs (20 weeks, 50 kg) were purchased from Manthei Hog Farm, LLC. |
|---|---|
| Wild animals | This study does not involve wild animals. |
| Field-collected samples | This study does not involve field-collected samples. |
| Ethics oversight | Animal procedures for rat and mice were reviewed and approved by the Massachusetts Institute of Technology Committee on Animal Care.<br><br>Animal procedures for pig were were reviewed and approved by the Mayo Clinic IACUC at Rochester. |

Note that full information on the approval of the study protocol must also be provided in the manuscript.

