## [Peer Review File · Nature]

Manuscript Title: Adhesive Anti-fibrotic Interfaces on Diverse Organs

Editorial Notes:

Redactions – unpublished data

Reviewer Comments & Author Rebuttals

Reviewer Reports on the Initial Version:

Referees' comments:

Referee #1 (Remarks to the Author):

A. Summary of the key results –

This paper documents a novel tissue adhesive that reduces the formation of a fibrous capsule at the tissue/material interface when implanted in the body. The FBR is a major limitation to the success of implantable devices. The authors go on to monitor the device/adhesive using a number of implant sites and implant periods and then show that the functional capacity of electrodes embedded within the adhesive is maintained for 28 days when compared to a non-adhesive patch. The loss of function in the non-adhesive group was due to the FBR and dislodgement of the device from the heart surface. This is really important as this demonstrates that this failure mode is overcome and that device function is maintained over time. They group completed extensive and rigorous tissue analysis and also verified tissue response using PCR for known FBR mediators and wider RNA-seq/omics to define the response and demonstrate the difference in a head to head comparison.

B. Originality and significance: if not novel, please include reference

This paper is significant and one of the first I have seen to show a lack of capsule formation and a reduced foreign body response to materials when placed directly on tissue surfaces. The adhesive nature of the material limits movement and removes a potential space for the fibrous capsule to form. The absence of friction and the material property matching to the soft tissues used for implant must also play a part in reducing the FBR. The paper and the rigorous approach to assessment will be influential to the field.

C. Data & methodology: validity of approach, quality of data, quality of presentation

The paper is well laid out and the study completed to a very high standard, the supplementary data provide all the information required to align to the claims made by the authors. The histology is excellent and detailed and the inclusion of further assessment of cellular response at the RNA level is important to understand the global response at the tissue/material interface. The final study to assess functional integration of a medical device and medical assessment using the method is strong – my only perceived limitation here is that the time-point is short and there is some drift in the signal. Albeit non-significant, this drift may be due to the lower n number in the study and the 28 time point chosen. An extended assessment of this outcome would be warranted. Furthermore, the scale up of this approach and the response of the material in larger animals may be warranted because the next step on clinical translation would be showing that the positive outcome is maintained when a large surface area is covered in large animal tissues – for instance, a 7 day study in a porcine model may be appropriate to

demonstrate the clinical scale possibility, an additional species etc.

B. The major limitation I see with the data as presented is the single rodent strain used for the full assessment and the overall low n numbers across the sample sets. We have seen many papers published over the last decade that have shown a reduced FBR in a single strain and then when translated to other rodent models or larger preclinical models the response is varied. This is even more important when thinking of clinical translation, as the most recent evidence of reduced FBR to modified alginate chemistries led by Sigilon commercially and developed by leaders in the biomaterials field was not evident when translated to humans during early stage clinical trials when evidence of fibrous encapsulation was seen in removed sample biomaterials. One option to better incorporate the human immune system is to test some of these approaches in humanized mouse or rat models, particularly the innate immune response to biomaterials – one of the main drivers of the outcome. The authors should discuss this in detail including the limitations of the animal models chosen in the paper and possibly add some additional assessment in a non-sprague dawley strain or humanized mouse model. Single gender animals were also used for all in vivo assessment – a justification is needed to explain this choice as there have been papers in the literature that discuss the gender differences in biomaterial responses.

D. Appropriate use of statistics and treatment of uncertainties

All statistics presented were clear and appropriate. The only query I had when reading and reviewing in detail was the n number for each treatment group at the start of the experiment and then how many animals were sacrificed at each timepoint and the n used for data presentation in each graphic/figure. This needs to be further clarified in the materials and methods, particularly when trying to understand the histology data presented and the image analyses provided versus the final assessment of functional outcomes.

E. Conclusions: robustness, validity, reliability

The conclusion is clear and measured with the data presented. A further conclusion point on the next steps for clinical translation is important.

F. Suggested improvements: experiments, data for possible revision –

please see points on preclinical studies above. That is the main suggested improvements as this is an excellent paper and if the data is reproducible in additional strains/species and eventually in the clinic, it will have major impact in the medical device field, particularly the integration of devices that need uninterrupted interfacing with organs/tissues.

G. References: appropriate credit to previous work?

All references are appropriate and the field and state of the art is well showcased.

H. Clarity and context: lucidity of abstract/summary, appropriateness of abstract, introduction and conclusions

This is a very clear paper, well written, well presented with an excellent outcome.

Referee #2 (Remarks to the Author):

This paper reports the adhesive implant-tissue interface, which potentially avoids fibrous capsule formation of implanted biomaterials. The study was comprehensive with good biological analyses.

However, novelty of the adhesive material is not convinced enough due to the lack of comparison with recent studies. In addition, detailed characterization of dynamic adhesive property should be required to highlight the novelty of synthesized adhesives such as time-course change in adhesion force when equipped with different soft-to-hard biomaterials, leading to the generalization of implant-tissue interface. Therefore, the reviewer recommends to submit the current manuscript to more specialized journals such as Biomaterials, Biomaterials Science, and ACS Applied Materials & Interfaces. The following comments would be useful tips upon submission.

[Comments]

1. Comparison with conventional adhesives (e.g., fibrin glues) as well as recently-reported adhesives should be introduced. For example, following papers should be considered:
<https://www.science.org/doi/10.1126/science.aah6362> <https://www.nature.com/articles/s41551-018-0261-7> <https://www.nature.com/articles/s41563-021-01051-x>
2. Flexural rigidity (in terms of Young's modulus and thickness) should be considered for investigating the implant-tissue interface. The present study mainly employed polyurethane, but other materials with different flexural rigidities should be tried.
3. The present study only focused on the interface between implant and tissue. However, fibrous capsule could be also propagated from the peripheries of the implant. How should we avoid such invasion totally?
4. Considering the clinical applications, large animal models (e.g., swine) should be examined since mechanical environment surrounding the adhesive implant is completely different from small animals. In addition, brain is also important body parts to be examined with bioelectrodes.

Referee #3 (Remarks to the Author):

This manuscript reports a strategy that an implant with an interfacial hydrogel adhesive can minimize foreign body reaction and fibrous capsule formation at the implant-tissue interface by conformal interfacial integration with the tissue. The finding is interesting and important to understand the foreign body reaction of host tissues to an implant, and the strategy shows values in material designs to minimize foreign body reaction and fibrosis. The data have been clearly presented and the manuscript is well written. However, there are several prior reports that have demonstrated a similar strategy to minimize foreign body reaction using a hydrogel layer or hydrogel adhesive between an implant and host tissues (e.g., Yang et al., Functional hydrogel interface materials for advanced bioelectronics devices, *Acc. Mater. Res.*, 2021 (Review)). Overall, the strategy is not completely new, but the studies and findings are undoubtedly valuable and influential. In addition, the current data appear difficult to support the claim of a long-term fibrous capsule-free performance. The major and minor issues are shown below.

1. The non-adhesive implant is sutured on the organ surfaces. The suture is non-degradable and present across the experimental time. Is it possible that the tissue injuries caused by suturing and the presence of suture material complicated the foreign body reaction cascade and contributed to induce a stronger inflammatory response and thus a thicker fibrous capsule for the non-adhesive implant compared to the

adhesive one? In Fig. 1 and 2, the skeletal muscle tissue is apparently loose in the non-adhesive implant-tissue samples compared to the adhesive implant-tissue samples. Is it likely because of tissue damages caused by suturing or other reasons? If both adhesive and non-adhesive implants are applied to an injured organ surface, can the adhesive implant still significantly prevent fibrous capsule formation compared to the non-adhesive implant?

2. In both Fig. 3 and Fig. 4, the standard deviations for some data points are relatively large. Although statistical analyses are performed to show significant differences for some data pairs, are the sample sizes ($n=3$ in Fig. 3g-i, and $n=9$ in Fig. 4c-f) appropriate to obtain scientifically valid data with a power level of 85% or above?

3. What are the mechanical properties of the adhesive and non-adhesive layers, and the two corresponding implants with the adhesive and non-adhesive layers, respectively? If the compositions of the hydrogel adhesive are changed, can this strategy similarly prevent the fibrous capsule formation? Because the composition changes will result in changes of crosslinking densities, adhesive performance, and mechanical properties of the hydrogel adhesive, and thus likely affect the conformal interfacial integration and host responses. This is important to know if this strategy is applicable to a wide range of scenarios, or just limited to the current composition.

4. Since the conformal interfacial integration is the key feature to prevent fibrous capsule formation as the authors claimed in this study, can this strategy be generalized to other polymeric hydrogel adhesives as the interfacial adhesive to prevent fibrous capsule formation?

5. Supplementary Fig. 4 only show histology images of the adhesive implant-abdominal wall on days 84 post-implantation. It appears there is a mild fibrous capsule at the interface which is different with the mesothelium tissue as the native tissue sample demonstrates. No other adhesive implant-organs are provided to illustrate the efficiencies to prevent fibrous capsule formation at this time point, given different mechanical properties and host responses of the different organs to the implant. Additionally, the electrodes with the adhesive interface were evaluated for only 28 days in a rat heart model (Fig. 6). Therefore, the current data appear difficult to support the claim of long-term fibrous capsule-free performance.

Minor issues:

1. In Methods section, what are the pH level, salt concentration, and temperature of the PBS bath used to prepare the non-adhesive implant?

2. Are the thicknesses of the adhesive and PU layers on the electrodes similar to the adhesive implant? It is better to add a picture of the designed electrodes in the supplementary data.

Author Rebuttals to Initial Comments:

Response to Referee #1

General comment. A. Summary of the key results –

This paper documents a novel tissue adhesive that reduces the formation of a fibrous capsule at the tissue/material interface when implanted in the body. The FBR is a major limitation to the success of implantable devices. The authors go on to monitor the device/adhesive using a number of implant sites and implant periods and then show that the functional capacity of electrodes embedded within the adhesive is maintained for 28 days when compared to a non-adhesive patch. The loss of function in the non-adhesive group was due to the FBR and dislodgement of the device from the heart surface. This is really important as this demonstrates that this failure mode is overcome and that device function is maintained over time. They group completed extensive and rigorous tissue analysis and also verified tissue response using PCR for known FBR mediators and wider RNA-seq/omics to define the response and demonstrate the difference in a head to head comparison.

B. Originality and significance: if not novel, please include reference

This paper is significant and one of the first I have seen to show a lack of capsule formation and a reduced foreign body response to materials when placed directly on tissue surfaces. The adhesive nature of the material limits movement and removes a potential space for the fibrous capsule to form. The absence of friction and the material property matching to the soft tissues used for implant must also play a part in reducing the FBR. The paper and the rigorous approach to assessment will be influential to the field.

Response. Thank you very much for your positive feedback on our work. To fully address the reviewers' comments and concerns, we have substantially revised the manuscript with newly added or revised 4 Main Figures, 6 Supplementary Figures, analyses, and clarifications. The newly added text in the revised manuscript is highlighted in blue color.

Comment 1. C. Data & methodology: validity of approach, quality of data, quality of presentation

The paper is well laid out and the study completed to a very high standard, the supplementary data provide all the information required to align to the claims made by the authors. The histology is excellent and detailed and the inclusion of further assessment of cellular response at the RNA level is important to understand the global response at the tissue/material interface. The final study to assess functional integration of a medical device and medical assessment using the method is strong – my only perceived limitation here is that the time-point is short and there is some drift in the signal. Albeit non-significant, this drift may be due to the lower n number in the study and the 28 time point chosen. An extended assessment of this outcome would be warranted. Furthermore, the scale up of this approach and the response of the material in larger animals may be warranted because the next step on clinical translation would be showing that the positive outcome is maintained when a large surface area is covered in large animal tissues – for instance, a 7 day study in a porcine model may be appropriate to demonstrate the clinical scale possibility, an additional species etc.

Response 1. Thank you for your insightful comment. Following the reviewer's suggestions, we added the following new data in the revised manuscript: 1) electrophysiological recording and stimulation of the heart for an extended period of up to 84 days, 2) pre-clinical studies in other species including a large animal model (porcine), immunocompetent C57BL/6 mice, and HuCD34-NCG humanized mice. These additional data clearly demonstrate that the proposed adhesive anti-fibrotic interface can provide a stable bioelectronic interface in the long-term and broad applicability to diverse species with human-like anatomical scales (porcine) as well as immune responses (humanized mice). To clarify these points, we have added the following figures and paragraphs in the revised manuscript:

On Page 5, "To test our hypothesis in diverse animal models, we implanted the adhesive and non-adhesive implants on the abdominal wall surface of immunocompetent C57BL/6 mice and HuCD34-NCG humanized mice (Fig. 5a,c). Note that immunocompetent C57BL/6 mice are known to produce fibrosis and foreign body reactions similar to those observed in human patients³², while HuCD34-NCG humanized mice provide human-like immune responses³³. Histological analysis shows that the adhesive implant-tissue interface exhibits no observable formation of the fibrous capsule, comparable to the native tissue on day 28 post-implantation for both C57BL/6 (Fig. 5b) and HuCD34-NCG (Fig. 5d)

mouse models. In contrast, the non-adhesive implant-tissue interface shows substantial formation of the fibrous capsule in both models (Fig. 5b,d).

To further test our hypothesis in human-scale anatomy, we implanted the adhesive and non-adhesive implants in porcine models (Fig. 5e and Supplementary Fig. 16a). Macroscopic observations demonstrate that the adhesive implant maintains stable integration with the surface of the porcine abdominal wall and small intestine on day 7 post-implantation in vivo (Supplementary Fig. 15). Histological analysis shows that the adhesive implant forms conformal integration with the tissue surface without observable formation of the fibrous capsule at the implant-tissue interface on day 7 post-implantation for both the abdominal wall (Fig. 5f) and small intestine (Supplementary Fig. 16b). In contrast, the non-adhesive implant-tissue interface exhibits substantial formation of the fibrous capsule (Fig. 5f and Supplementary Fig. 16g), in agreement with the observations in the rodent models.”

Fig. R1 (added as Fig. 5) | Adhesive anti-fibrotic interfaces in diverse animal models. **a**, Schematic illustration for the study design based on the C57BL/6 mouse model. **b**, Representative histology images stained with Masson’s trichrome (MTS) and hematoxylin and eosin (H&E) for native tissue (left), adhesive implant (middle), and non-adhesive implant (right) collected on 28 days post-implantation on the abdominal wall. **c**, Schematic illustration for the study design based on the HuCD34-NCG humanized mouse model. **d**, Representative histology images stained with MTS and H&E for native tissue (left), adhesive implant (middle), and non-adhesive implant (right) collected on 28 days post-implantation on the abdominal wall. **e**, Schematic illustration for the study design based on the porcine model. **f**, Representative histology images stained with MTS and H&E for native tissue (left), adhesive implant (middle), and non-adhesive implant (right) collected on 7 days post-implantation on the abdominal wall. Black dotted lines in images indicate the implant-tissue interface; yellow dotted lines in images indicate the mesothelium-fibrous capsule interface. Parts of **a,c,e** were created with BioRender.com.

On Page 5, “To explore the potential utility of the adhesive anti-fibrotic interfaces, we demonstrated long-term in vivo electrophysiological recording and stimulation enabled by the implantable electrodes with the adhesive interface in a rat model for 84 days (Fig. 6). For continuous in vivo monitoring and modulation of the electrocardiogram, electrodes with the adhesive interface or the non-adhesive interface were implanted on the epicardial surface of animals for electrophysiological recording and stimulation on days 0, 3, 7, 14, 28, 56, and 84 post-implantation (Fig. 6a and Supplementary Fig. 17).

Macroscopic observations showed that the electrodes with the adhesive interface maintained stable integration with the heart after 84 days of implantation *in vivo* (Fig. 6b). The amplitude of the R-wave recorded by the electrodes with the adhesive interface was consistently maintained throughout the study duration (84 days, Fig. 6e-g), whereas the R-wave amplitude recorded by the electrodes with the non-adhesive interface exhibited a substantial decrease over time (Fig. 6e). For electrophysiological stimulation, the minimal stimulation current pulse amplitude needed to successfully pace the heart gradually increased until 7 days post-implantation and eventually failed to pace the heart 28 days post-implantation for the electrodes with the non-adhesive interface (Fig. 6c). In contrast, the electrodes with the adhesive interface exhibited a consistent minimal stimulation current pulse amplitude for pacing and successfully maintained the capability to pace the heart for the duration of the study (84 days, Fig. 6d). These results are consistent with the histological findings from the tissues collected on day 28 post-implantation, where the electrodes with the non-adhesive interface showed encapsulation and physical separation from the epicardial surface by a thick fibrous capsule (Fig. 6h). In contrast, the electrodes with the adhesive interface showed conformal contact with the epicardial surface without observable formation of the fibrous capsule (Fig. 6i).”

Fig. R2 (added as Fig. 6) | Long-term in vivo bi-directional electrical communication via the adhesive anti-fibrotic interfaces. **a**, Schematic illustrations for in vivo electrophysiological recording and stimulation via implanted electrodes with the non-adhesive or the adhesive implant-tissue interfaces. **b**, Photographs of the heart collected on 0 and 84 days post-implantation for electrodes with the adhesive interface. White dotted lines in photographs indicate the boundary of implants. **c**, Representative epicardial electrocardiograms after stimulation via implanted electrodes with the non-adhesive implant-tissue interface on 0, 3, 7, 14, and 28 days post-implantation on a rat heart. **d**, Representative epicardial electrocardiograms after stimulation via implanted electrodes with the adhesive implant-tissue interface on 0, 14, 28, 56, and 84 days post-implantation on a rat heart. **e-g**, Recorded R-wave amplitude via implanted electrodes with the non-adhesive (black) and the adhesive (red) implant-tissue interfaces on day 28 (**e**), day 56 (**f**), and day 84 (**g**) post-implantation on a rat heart. Inset plots show representative recorded waveforms. **h, i**, Representative histology images stained with Masson's trichrome (MTS, left) and hematoxylin and eosin (H&E, right) of the electrodes with non-adhesive (**h**) and the adhesive (**i**) implant collected on 28 days post-implantation on a rat heart. * in images indicates the implant; yellow dotted lines in images indicate the implant-tissue interface. Values in **e-g** represent the mean and the standard deviation ($n = 6$; independent samples). Statistical significance and P values are determined by two-sided unpaired t -tests; ns, not significant.

Comment 2. B. The major limitation I see with the data as presented is the single rodent strain used for the full assessment and the overall low n numbers across the sample sets. We have seen many

papers published over the last decade that have shown a reduced FBR in a single strain and then when translated to other rodent models or larger preclinical models the response is varied. This is even more important when thinking of clinical translation, as the most recent evidence of reduced FBR to modified alginate chemistries led by Sigilon commercially and developed by leaders in the biomaterials field was not evident when translated to humans during early stage clinical trials when evidence of fibrous encapsulation was seen in removed sample biomaterials. One option to better incorporate the human immune system is to test some of these approaches in humanized mouse or rat models, particularly the innate immune response to biomaterials – one of the main drivers of the outcome. The authors should discuss this in detail including the limitations of the animal models chosen in the paper and possibly add some additional assessment in a non-sprague dawley strain or humanized mouse model. Single gender animals were also used for all in vivo assessment – a justification is needed to explain this choice as there have been papers in the literature that discuss the gender differences in biomaterial responses.

Response 2. Thank you for your insightful comment. Following the reviewer’s suggestions, we added the following new data in the revised manuscript: 1) Additional numbers of animals for pre-clinical evaluations in both sexes (female and male), 2) pre-clinical studies in other species including a large animal model (porcine), immunocompetent C57BL/6 mice, and HuCD34-NCG humanized mice. These additional data clearly demonstrate the broad applicability of the proposed work in diverse species and sexes with human-like anatomical scales (porcine) and immune responses (humanized mice). To clarify these points, we have added the following figures (**Fig. R1** in **Response 1**) and paragraphs in the revised manuscript:

On Page 5, “To test our hypothesis in diverse animal models, we implanted the adhesive and non-adhesive implants on the abdominal wall surface of immunocompetent C57BL/6 mice and HuCD34-NCG humanized mice (**Fig. 5a,c**). Note that immunocompetent C57BL/6 mice are known to produce fibrosis and foreign body reactions similar to those observed in human patients³², while HuCD34-NCG humanized mice provide human-like immune responses³³. Histological analysis shows that the adhesive implant-tissue interface exhibits no observable formation of the fibrous capsule, comparable to the native tissue on day 28 post-implantation for both C57BL/6 (**Fig. 5b**) and HuCD34-NCG (**Fig. 5d**) mouse models. In contrast, the non-adhesive implant-tissue interface shows substantial formation of the fibrous capsule in both models (**Fig. 5b,d**).

To further test our hypothesis in human-scale anatomy, we implanted the adhesive and non-adhesive implants in porcine models (**Fig. 5e** and **Supplementary Fig. 16a**). Macroscopic observations demonstrate that the adhesive implant maintains stable integration with the surface of the porcine abdominal wall and small intestine on day 7 post-implantation in vivo (**Supplementary Fig. 15**). Histological analysis shows that the adhesive implant forms conformal integration with the tissue surface without observable formation of the fibrous capsule at the implant-tissue interface on day 7 post-implantation for both the abdominal wall (**Fig. 5f**) and small intestine (**Supplementary Fig. 16b**). In contrast, the non-adhesive implant-tissue interface exhibits substantial formation of the fibrous capsule (**Fig. 5f** and **Supplementary Fig. 16g**), in agreement with the observations in the rodent models.”

Comment 3. D. Appropriate use of statistics and treatment of uncertainties

All statistics presented were clear and appropriate. The only query I had when reading and reviewing in detail was the n number for each treatment group at the start of the experiment and then how many animals were sacrificed at each timepoint and the n used for data presentation in each graphic/figure. This needs to be further clarified in the materials and methods, particularly when trying to understand the histology data presented and the image analyses provided versus the final assessment of functional outcomes.

Response 3. Thank you for your insightful comment. We have added detailed information about the number of animals tested for each study group and time points in the Methods section and figure captions of the revised manuscript. Also, the Reporting Summary has been updated accordingly in the revised manuscript.

Comment 4. E. Conclusions: robustness, validity, reliability

The conclusion is clear and measured with the data presented. A further conclusion point on the next steps for clinical translation is important.

F. Suggested improvements: experiments, data for possible revision –

please see points on preclinical studies above. That is the main suggested improvements as this is an excellent paper and if the data is reproducible in additional strains/species and eventually in the clinic, it will have major impact in the medical device field, particularly the integration of devices that need uninterrupted interfacing with organs/tissues.

G. References: appropriate credit to previous work?

All references are appropriate and the field and state of the art is well showcased.

H. Clarity and context: lucidity of abstract/summary, appropriateness of abstract, introduction and conclusions

This is a very clear paper, well written, well presented with an excellent outcome.

Response 4. Thank you for your comments and valuable insights. We have reflected the reviewer's suggestions in the revised manuscript including additional pre-clinical studies in diverse and clinically-relevant species (please see **Response 2**).

We greatly appreciate the reviewer's time and effort to provide insightful comments and suggestions. We hope that our revised manuscript and responses address all of your concerns about the work.

Response to Referee #2

General Comment. This paper reports the adhesive implant-tissue interface, which potentially avoids fibrous capsule formation of implanted biomaterials. The study was comprehensive with good biological analyses. However, novelty of the adhesive material is not convinced enough due to the lack of comparison with recent studies. In addition, detailed characterization of dynamic adhesive property should be required to highlight the novelty of synthesized adhesives such as time-course change in adhesion force when equipped with different soft-to-hard biomaterials, leading to the generalization of implant-tissue interface. Therefore, the reviewer recommends to submit the current manuscript to more specialized journals such as Biomaterials, Biomaterials Science, and ACS Applied Materials & Interfaces. The following comments would be useful tips upon submission.

Response. Thank you for your constructive feedback on our work. To fully address the reviewers' comments and concerns, we have substantially revised the manuscript with newly added or revised 4 Main Figures, 6 Supplementary Figures, analyses, and clarifications. The newly added text in the revised manuscript is highlighted in blue color. In the following detailed responses, we have clarified the novelty of the current work over the previously reported anti-fibrotic strategies and adhesives as well as the general applicability of the proposed strategies based on diverse and clinically-relevant animal models.

Comment 1. Comparison with conventional adhesives (e.g., fibrin glues) as well as recently-reported adhesives should be introduced. For example, following papers should be considered: <https://www.science.org/doi/10.1126/science.aah6362> <https://www.nature.com/articles/s41551-018-0261-7><https://www.nature.com/articles/s41563-021-01051-x>

Response 1. Thank you for suggesting the relevant papers in the literature. The suggested works have been cited in the revised manuscript (as Refs. 6, 7, and 28). We would like to emphasize that the current work does not aim to develop adhesive biomaterial or compare adhesion performance to the existing adhesives. Rather, the current work's goal is to report and validate the anti-fibrotic adhesive interface as a new strategy to prevent substantial formation of the fibrous capsule.

To our best knowledge, including the two previous works suggested by the reviewer, no previous study demonstrated 1) an adhesive interface to achieve anti-fibrotic implant-tissue interface in long-term, 2) the adhesive anti-fibrotic interface generally applicable to diverse organs and species (including clinically-relevant models like porcine and humanized mice), and 3) systematic and comprehensive analyses of the adhesive anti-fibrotic interface based on the biomaterial in both adhesive and non-adhesive states without chemical/compositional differences.

These features clearly highlight the novelty of the current work over recently reported adhesives and previously reported anti-fibrotic strategies in the field. Furthermore, to address the reviewer's concern, we have added a comparison with commercially-available adhesives such as synthetic adhesives (Coseal) and fibrin-based adhesives (Tisseel). To clarify these points, we have added the following figures and paragraphs in the revised manuscript:

On Page 3, "To investigate the effect of adhesive interfaces with varying compositions and properties, we replaced the PVA-based adhesive interface with a chitosan-based adhesive interface²² (see Methods for the preparation of the chitosan-based adhesive interface). Compared to the PVA-based adhesive interface, the chitosan-based adhesive interface offers a different composition and Young's modulus, yet it demonstrates comparable adhesion performance (**Supplementary Fig. 8a-d**). Histological analysis shows that the chitosan-based adhesive interface exhibits no observable formation of the fibrous capsule 14 days post-implantation (**Supplementary Fig. 8e,f**). Notably, the implants adhered to the abdominal wall surface using commercially-available tissue adhesives including Coseal and Tisseel show the substantial formation of the fibrous capsule 14 days post-implantation (**Supplementary Fig. 9**). This may be attributed to unstable long-term adhesion of the commercially-available tissue adhesives with the tissue surface *in vivo*²⁶."

Fig. R3 (added as Supplementary Fig. 9) | Adhesive interface by commercially-available tissue adhesives. a,b, Representative histology images stained with Masson's trichrome (left) and hematoxylin and eosin (right) for the implant integrated to the abdominal wall surface by Coseal (a) and Tisseel (b) collected on day 14 post-implantation. * in images indicates the implant; black dotted lines indicate the implant-tissue interface; yellow dotted lines indicate the mesothelium-fibrous capsule interface. SM, skeletal muscle; FC, fibrous capsule.

Comment 2. Flexural rigidity (in terms of Young's modulus and thickness) should be considered for investigating the implant-tissue interface. The present study mainly employed polyurethane, but other materials with different flexural rigidities should be tried.

Response 2. Thank you for your insightful comment. In the current study, two implant materials (polyurethane and gold electrode) were adopted and tested in vivo for the long-term adhesive anti-fibrotic interface whose Young's moduli and thickness were provided in the revised manuscript (**Supplementary Table 2**). As shown in the table, two implant materials have comparable thickness while having over three orders of magnitudes difference in Young's moduli. As flexural rigidity (or bending stiffness) is proportional to Young's modulus for the given thickness, the wide differences in Young's moduli of the implants used in the study demonstrate the robustness of the proposed strategy.

Table R1 (added as Supplementary Table 2) | Mechanical properties of the implants in this study

Implant	Young's modulus	Thickness
Polyurethane	2.1 MPa	100 μm
Gold electrode	79 GPa	50 μm

Comment 3. The present study only focused on the interface between implant and tissue. However, fibrous capsule could be also propagated from the peripheries of the implant. How should we avoid such invasion totally?

Response 3. As demonstrated in **Fig. R4** (Supplementary Fig. 3 in the revised manuscript), we have shown that the adhesive anti-fibrotic interface is maintained across the entire adhesive interface including the peripheries of the implant.

Fig. R4 (added as Supplementary Fig. 3) | Adhesive implant histology. **a**, Representative histology images stained with Masson's trichrome (MTS, left) and hematoxylin and eosin (H&E, right) of the adhesive implant collected on day 28 post-implantation to the abdominal wall. Black and red dotted areas indicate the implant-tissue interface and the implant-abdominal cavity interface, respectively. **b,c**, Representative histology images stained with MTS (left) and H&E (right) of the implant-tissue interface (b) and implant-cavity interface (c) for the adhesive implant collected on day 28 post-implantation to the abdominal wall.

Comment 4. Considering the clinical applications, large animal models (e.g., swine) should be examined since mechanical environment surrounding the adhesive implant is completely different from small animals. In addition, brain is also important body parts to be examined with bioelectrodes.

Response 4. Following the reviewer's suggestions, we added new pre-clinical studies in other species including a large animal model (porcine), immunocompetent C57BL/6 mice, and HuCD34-NCG humanized mice. These additional data clearly demonstrate the broad applicability of the proposed work in clinically-relevant animal models with human-like anatomical scales (porcine) and immune responses (humanized mice). The brain was not examined in the current study due to distinctive biological characteristics and foreign body reaction of the brain compared to other organs, which can be future research direction. To clarify these points, we have added the following figures and paragraphs in the revised manuscript:

On Page 5, "To test our hypothesis in diverse animal models, we implanted the adhesive and non-adhesive implants on the abdominal wall surface of immunocompetent C57BL/6 mice and HuCD34-NCG humanized mice (Fig. 5a,c). Note that immunocompetent C57BL/6 mice are known to produce fibrosis and foreign body reactions similar to those observed in human patients³², while HuCD34-NCG humanized mice provide human-like immune responses³³. Histological analysis shows that the adhesive implant-tissue interface exhibits no observable formation of the fibrous capsule, comparable to the native tissue on day 28 post-implantation for both C57BL/6 (Fig. 5b) and HuCD34-NCG (Fig. 5d) mouse models. In contrast, the non-adhesive implant-tissue interface shows substantial formation of the fibrous capsule in both models (Fig. 5b,d).

To further test our hypothesis in human-scale anatomy, we implanted the adhesive and non-adhesive implants in porcine models (Fig. 5e and Supplementary Fig. 16a). Macroscopic observations demonstrate that the adhesive implant maintains stable integration with the surface of the porcine abdominal wall and small intestine on day 7 post-implantation in vivo (Supplementary Fig. 15). Histological analysis shows that the adhesive implant forms conformal integration with the tissue surface without observable formation of the fibrous capsule at the implant-tissue interface on day 7 post-implantation for both the abdominal wall (Fig. 5f) and small intestine (Supplementary Fig. 16b). In contrast, the non-adhesive implant-tissue interface exhibits substantial formation of the fibrous capsule (Fig. 5f and Supplementary Fig. 16g), in agreement with the observations in the rodent models."

Fig. R5 (added as Fig. 5) | Adhesive anti-fibrotic interfaces in diverse animal models. **a**, Schematic illustration for the study design based on the C57BL/6 mouse model. **b**, Representative histology images stained with Masson's trichrome (MTS) and hematoxylin and eosin (H&E) for native tissue (left), adhesive implant (middle), and non-adhesive implant (right) collected on 28 days post-implantation on the abdominal wall. **c**, Schematic illustration for the study design based on the HuCD34-NCG humanized mouse model. **d**, Representative histology images stained with MTS and H&E for native tissue (left), adhesive implant (middle), and non-adhesive implant (right) collected on 28 days post-implantation on the abdominal wall. **e**, Schematic illustration for the study design based on the porcine model. **f**, Representative histology images stained with MTS and H&E for native tissue (left), adhesive implant (middle), and non-adhesive implant (right) collected on 7 days post-implantation on the abdominal wall. Black dotted lines in images indicate the implant-tissue interface; yellow dotted lines in images indicate the mesothelium-fibrous capsule interface. Parts of **a,c,e** were created with BioRender.com.

Fig. R6 (added as Supplementary Fig. 16) | Adhesive anti-fibrotic interfaces in porcine model. **a**, Schematic illustration for the study design based on the porcine model. **b**, Representative histology images stained with Masson's trichrome (MTS) and hematoxylin and eosin (H&E) for native tissue (left), adhesive implant (middle), and non-adhesive implant (right) collected on 7 days post-implantation to the small intestine. Black dotted lines in images indicate the implant-tissue interface; yellow dotted lines in images indicate the mesothelium-fibrous capsule interface. Parts of **a** were created with BioRender.com.

We greatly appreciate the reviewer's time and effort to provide insightful comments and suggestions. We hope that our revised manuscript and responses address all of your concerns about the work.

Response to Referee #3

General Comment. This manuscript reports a strategy that an implant with an interfacial hydrogel adhesive can minimize foreign body reaction and fibrous capsule formation at the implant-tissue interface by conformal interfacial integration with the tissue. The finding is interesting and important to understand the foreign body reaction of host tissues to an implant, and the strategy shows values in material designs to minimize foreign body reaction and fibrosis. The data have been clearly presented and the manuscript is well written. However, there are several prior reports that have demonstrated a similar strategy to minimize foreign body reaction using a hydrogel layer or hydrogel adhesive between an implant and host tissues (e.g., Yang et al., Functional hydrogel interface materials for advanced bioelectronics devices, *Acc. Mater. Res.*, 2021 (Review)). Overall, the strategy is not completely new, but the studies and findings are undoubtedly valuable and influential. In addition, the current data appear difficult to support the claim of a long-term fibrous capsule-free performance. The major and minor issues are shown below.

Response. Thank you very much for your positive feedback on our work. To fully address the reviewers' comments and concerns, we have substantially revised the manuscript with newly added or revised 4 Main Figures, 6 Supplementary Figures, analyses, and clarifications. The newly added text in the revised manuscript is highlighted in blue color. In the following detailed responses, we have clarified the novelty of the current work over the previously reported anti-fibrotic strategies and adhesives as well as the general applicability of the proposed strategies based on diverse and clinically-relevant animal models.

Comment 1. The non-adhesive implant is sutured on the organ surfaces. The suture is non-degradable and present across the experimental time. Is it possible that the tissue injuries caused by suturing and the presence of suture material complicated the foreign body reaction cascade and contributed to induce a stronger inflammatory response and thus a thicker fibrous capsule for the non-adhesive implant compared to the adhesive one? In Fig. 1 and 2, the skeletal muscle tissue is apparently loose in the non-adhesive implant-tissue samples compared to the adhesive implant-tissue samples. Is it likely because of tissue damages caused by suturing or other reasons? If both adhesive and non-adhesive implants are applied to an injured organ surface, can the adhesive implant still significantly prevent fibrous capsule formation compared to the non-adhesive implant?

Response 1. Thank you for your insightful comment. Following the reviewer's insight, we have added new data to investigate the suture's effect on the adhesive implant in the revised manuscript. Our new data clearly demonstrates that the adhesive implant-tissue interface can avoid excessive formation of the fibrous capsule even when sutures are introduced at the corners of the samples whereas the suture point shows fibrosis due to the disruption of the adhesive implant-tissue interface. To clarify these points, we have added the following figure and paragraph in the revised manuscript:

R7 (added as Supplementary Fig. 7) | Adhesive implant-tissue interface with sutures. **a**, Schematic illustrations of the adhesive implant with sutures at the corners. **b**, Representative histology image stained with hematoxylin and eosin (H&E) for the adhesive implant with sutures on the abdominal wall collected on day 28 post-implantation. **c,d**, Representative histology images stained with Masson's trichrome (MTS, left) and H&E (right) for the suture point (c) and the intact adhesive-tissue interface (d) collected on day 28 post-implantation to the abdominal wall. * in images indicates the implant; black dotted lines indicate the implant-tissue interface.

On Page 3, "Unlike the adhesive-tissue interface, the mock device-cavity interface of the adhesive implant undergoes fibrous capsule formation similar to the non-adhesive implant (**Supplementary Fig. 3**). To investigate the potential influence of suture-induced tissue damage, sutures were introduced to the corners of the adhesive implant, similar to the non-adhesive implant (**Supplementary Fig. 7a**). The histological analysis shows that the suture point exhibits the formation of fibrosis (**Supplementary Fig. 7b, c**), but the intact adhesive-tissue interface demonstrates no observable formation of the fibrotic capsule (**Supplementary Fig. 7b,d**). Collectively, these data further confirm that the adhesive interface is required to prevent the observable formation of the fibrous capsule."

Comment 2. In both Fig. 3 and Fig. 4, the standard deviations for some data points are relatively large. Although statistical analyses are performed to show significant differences for some data pairs, are the sample sizes (n=3 in Fig. 3g-i, and n=9 in Fig. 4c-f) appropriate to obtain scientifically valid data with a power level of 85% or above?

Response 2. Thank you for your insightful comment. We have performed post-hoc power analysis on these datasets (by using G*Powder 3.1) to assess the overall power and appropriateness of the sample size in Figs. 3 and 4 (summarized in **Tables R2** and **R3** below). The post-hoc power analysis shows that the majority achieves the power of 85% or above, supporting the appropriateness of the sample size used in the current study. Even though some genes like *Nos2* and *S100a8* display slightly lower power values when assessed via RT-qPCR, we have made sure in our comprehensive characterization of the Fibrosis mechanism to include multiple complementary modalities to measure differences in key cell types and pathways of interest. As deviations between transcriptome and proteome are common, we evaluated *Nos2* at the protein level (i.e., iNOS) and show clear highly significant differences with immunofluorescent staining (Fig. 3 and Supplementary Fig. 12 in the revised manuscript). We also want to note that *Nos2* is expressed by a multitude of cell types which could contribute to the high standard deviations observed. Similarly, with *S100a8* which was selected as a neutrophil marker at the transcriptome, we utilized Neutrophil elastase at the protein level to show unequivocally diminished numbers of neutrophils (Fig. 3 in the revised manuscript). We are confident that our presented extensive analyses with RNA-seq and RT-qPCR, as well as immunofluorescent staining and Luminex protein quantitation provide strong evidence to support the anti-fibrotic effect of the adhesive-implant.

[Redacted]

Comment 3. What are the mechanical properties of the adhesive and non-adhesive layers, and the two corresponding implants with the adhesive and non-adhesive layers, respectively?

If the compositions of the hydrogel adhesive are changed, can this strategy similarly prevent the fibrous capsule formation?

Because the composition changes will result in changes of crosslinking densities, adhesive performance, and mechanical properties of the hydrogel adhesive, and thus likely affect the conformal interfacial integration and host responses. This is important to know if this strategy is applicable to a wide range of scenarios, or just limited to the current composition.

Response 3. Thank you for your insightful comment. Following the reviewer’s insightful suggestions, we have added the following new data in the revised manuscript: 1) an adhesive interface with varying composition and mechanical properties and 2) mechanical characterizations of the adhesive interface (Young’s modulus, interfacial toughness, shear strength, tensile strength). These new data demonstrate that adhesive interfaces with varying composition and mechanical properties can achieve adhesive anti-fibrotic interfaces when they provide comparable adhesive performance. To clarify these points, we have added the following figure and paragraph in the revised manuscript:

Fig. R8 (added as Supplementary Fig. 8) | Chitosan-based adhesive interface. **a**, Engineering stress vs. stretch curves for the PVA-based and chitosan-based adhesive interface interfaces. E_{PVA} , Young’s modulus of the PVA-based adhesive interface; $E_{Chitosan}$, Young’s modulus of the chitosan-based adhesive interface. **b-d**, Interfacial toughness (b), shear strength (c), and tensile strength (d) of the PVA-based and chitosan-based adhesive interfaces on ex vivo porcine skin. **e,f**, Representative histology images stained with Masson’s trichrome (MTS) and hematoxylin and eosin (H&E) for native tissue (left), the adhesive implant (middle), and non-adhesive implant (right) collected on day 14 post-

implantation to the abdominal wall based on the PVA-based adhesive interface (e) and the chitosan-based adhesive interface (f). Black and yellow dotted lines in the images indicate the implant-tissue interface and the mesothelium-fibrous capsule interface, respectively. Values in **b-d** represent the mean and the standard deviation ($n = 3$; independent samples).

On Page 3,” To investigate the effect of adhesive interfaces with varying compositions and properties, we replaced the PVA-based adhesive interface with a chitosan-based adhesive interface²² (see Methods for the preparation of the chitosan-based adhesive interface). Compared to the PVA-based adhesive interface, the chitosan-based adhesive interface offers a different composition and Young's modulus, yet it demonstrates comparable adhesion performance (**Supplementary Fig. 8a-d**). Histological analysis shows that the chitosan-based adhesive interface exhibits no observable formation of the fibrous capsule 14 days post-implantation (**Supplementary Fig. 8e,f**). Notably, the implants adhered to the abdominal wall surface using commercially-available tissue adhesives including Coseal and Tisseel show the substantial formation of the fibrous capsule 14 days post-implantation (**Supplementary Fig. 9**). This may be attributed to unstable long-term adhesion of the commercially-available tissue adhesives with the tissue surface *in vivo*²⁶.”

We also wish to clarify that the adhesive and non-adhesive interfaces used in the current study have identical chemical composition and mechanical properties in the swollen state (equilibrium state in the physiological environment). This approach was adopted in the current study to minimize the potential variations in implant-tissue interactions other than the adhesive interface originating from the chemical and mechanical property differences between groups. To clarify these points, we have added the following paragraph in the revised manuscript:

On Page 2, “To test our hypothesis, we prepared an adhesive implant consisting of a mock device (polyurethane) and an adhesive layer^{22,23} consisting of interpenetrating networks between the covalently-crosslinked poly(acrylic acid) *N*-hydroxysuccinimide (NHS) ester and physically-crosslinked poly(vinyl alcohol) (PVA) (**Fig. 1c**). The adhesive layer provides highly conformal and stable integration of the implant with wet tissues²²⁻²⁴ (**Supplementary Fig. 1**). We further prepared a non-adhesive implant by fully swelling the same mock device and adhesive layer in a phosphate-buffered saline (PBS) bath before implantation. By swelling the implant in PBS, we removed its adhesive property²⁵ while keeping its chemical composition identical (see Methods for the preparation of the non-adhesive implant).”

Comment 4. Since the conformal interfacial integration is the key feature to prevent fibrous capsule formation as the authors claimed in this study, can this strategy be generalized to other polymeric hydrogel adhesives as the interfacial adhesive to prevent fibrous capsule formation?

Response 4. Please find the above **Response 3**.

Comment 5. Supplementary Fig. 4 only show histology images of the adhesive implant-abdominal wall on days 84 post-implantation. It appears there is a mild fibrous capsule at the interface which is different with the mesothelium tissue as the native tissue sample demonstrates.

No other adhesive implant-organs are provided to illustrate the efficiencies to prevent fibrous capsule formation at this time point, given different mechanical properties and host responses of the different organs to the implant.

Additionally, the electrodes with the adhesive interface were evaluated for only 28 days in a rat heart model (Fig. 6). Therefore, the current data appear difficult to support the claim of long-term fibrous capsule-free performance.

Response 5. Thank you for your insightful comment. Following the reviewer's suggestions, we have added the following new data in the revised manuscript: 1) long-term data for diverse organs for an extended period of up to 84 days and 2) long-term electrophysiological recording and stimulation of the heart for an extended period of up to 84 days. These newly added data clearly demonstrate that the

adhesive anti-fibrotic interface can provide a stable and functional long-term implant-tissue interface without substantial formation of the fibrous capsule for diverse organs. To clarify these points, we have added the following figures and paragraphs in the revised manuscript:

Fig. R9 (added as Fig. 1) | Adhesive anti-fibrotic interfaces. **a,b**, Schematic illustrations of a non-adhesive implant consisting of a mock device (polyurethane) and a non-adhesive layer (a) and long-term in vivo implantation with fibrous capsule formation at the implant-tissue interface (b). **c,d**, Schematic illustrations of an adhesive implant consisting of the mock device (polyurethane) and an adhesive layer (c) and long-term in vivo implantation without substantial fibrous capsule formation at the implant-tissue interface (d). **e-i**, Representative histology images stained with Masson's trichrome (MTS) and hematoxylin and eosin (H&E) for native tissue (left), the adhesive implant (middle), and non-adhesive implant (right) collected on day 84 post-implantation on the abdominal wall (e), colon (f), stomach (g), lung (h), and heart (i). Black and yellow dotted lines in the images indicate the implant-tissue interface and the mesothelium-fibrous capsule interface, respectively.

On Page 2, “Histological evaluation by a blinded pathologist indicates that the adhesive implant forms conformal integration with the organ surface (i.e., mesothelium) and shows no observable formation of the fibrous capsule on 28 days and up to 84 days post-implantation for diverse organs, including the abdominal wall, colon, stomach, lung, and heart (Fig. 1e-i, Supplementary Figs. 3 and 4). Furthermore, a transmission electron microscope (TEM) image of the adhesive implant-tissue interface shows that the adhesive layer maintains highly conformal and gapless integration with the collagenous layer of the mesothelium on a subcellular scale on 28 days post-implantation (Supplementary Fig. 5). In contrast, the non-adhesive implant undergoes substantial formation of the fibrous capsule at the implant-tissue interface for all organs, consistent with the foreign body reaction to the mock device alone (Fig. 1, Supplementary Figs. 4 and 6).”

Fig. R10 (added as Fig. 6) | Long-term in vivo bi-directional electrical communication via the adhesive anti-fibrotic interfaces. **a**, Schematic illustrations for in vivo electrophysiological recording and stimulation via implanted electrodes with the non-adhesive or the adhesive implant-tissue interfaces. **b**, Photographs of the heart collected on 0 and 84 days post-implantation for electrodes with the adhesive interface. White dotted lines in photographs indicate the boundary of implants. **c**, Representative epicardial electrocardiograms after stimulation via implanted electrodes with the non-adhesive implant-tissue interface on 0, 3, 7, 14, and 28 days post-implantation on a rat heart. **d**, Representative epicardial electrocardiograms after stimulation via implanted electrodes with the adhesive implant-tissue interface on 0, 14, 28, 56, and 84 days post-implantation on a rat heart. **e-g**, Recorded R-wave amplitude via implanted electrodes with the non-adhesive (black) and the adhesive (red) implant-tissue interfaces on day 28 (e), day 56 (f), and day 84 (g) post-implantation on a rat heart. Inset plots show representative recorded waveforms. **h,i**, Representative histology images stained with Masson's trichrome (MTS, left) and hematoxylin and eosin (H&E, right) of the electrodes with non-adhesive (h) and the adhesive (i) implant collected on 28 days post-implantation on a rat heart. * in images indicates the implant; yellow dotted lines in images indicate the implant-tissue interface. Values

in **e-g** represent the mean and the standard deviation ($n = 6$; independent samples) Statistical significance and \$P\$ values are determined by two-sided unpaired \$t\$ -tests; ns, not significant.

On Page 5, “To explore the potential utility of the adhesive anti-fibrotic interfaces, we demonstrated long-term in vivo electrophysiological recording and stimulation enabled by the implantable electrodes with the adhesive interface in a rat model for 84 days (Fig. 6). For continuous in vivo monitoring and modulation of the electrocardiogram, electrodes with the adhesive interface or the non-adhesive interface were implanted on the epicardial surface of animals for electrophysiological recording and stimulation on days 0, 3, 7, 14, 28, 56, and 84 post-implantation (Fig. 6a and Supplementary Fig. 17). Macroscopic observations showed that the electrodes with the adhesive interface maintained stable integration with the heart after 84 days of implantation in vivo (Fig. 6b). The amplitude of the R-wave recorded by the electrodes with the adhesive interface was consistently maintained throughout the study duration (84 days, Fig. 6e-g), whereas the R-wave amplitude recorded by the electrodes with the non-adhesive interface exhibited a substantial decrease over time (Fig. 6e). For electrophysiological stimulation, the minimal stimulation current pulse amplitude needed to successfully pace the heart gradually increased until 7 days post-implantation and eventually failed to pace the heart 28 days post-implantation for the electrodes with the non-adhesive interface (Fig. 6c). In contrast, the electrodes with the adhesive interface exhibited a consistent minimal stimulation current pulse amplitude for pacing and successfully maintained the capability to pace the heart for the duration of the study (84 days, Fig. 6d). These results are consistent with the histological findings from the tissues collected on day 28 post-implantation, where the electrodes with the non-adhesive interface showed encapsulation and physical separation from the epicardial surface by a thick fibrous capsule (Fig. 6h). In contrast, the electrodes with the adhesive interface showed conformal contact with the epicardial surface without observable formation of the fibrous capsule (Fig. 6i).”

Comment 6. In Methods section, what are the pH level, salt concentration, and temperature of the PBS bath used to prepare the non-adhesive implant?

Response 6. Thank you for your insightful comment. We have added the detailed condition and method for the preparation of the non-adhesive implant in the Methods section of the revised manuscript.

On Page 7, “Preparation of non-adhesive implants. To prepare the non-adhesive implant, the adhesive implant was immersed in a sterile 1X PBS (pH 7.4, 144 mg L⁻¹ potassium phosphate monobasic, 9,000 mg L⁻¹ sodium chloride, and 795 mg L⁻¹ sodium phosphate dibasic) bath at room temperature overnight. During this process, the adhesive layer of the implant reached the equilibrium swollen state and became non-adhesive by losing the capability to form physical (hydrogen bonds) and covalent (amide bonds) crosslinking with tissues²⁵.”

Comment 7. Are the thicknesses of the adhesive and PU layers on the electrodes similar to the adhesive implant? It is better to add a picture of the designed electrodes in the supplementary data.

Response 7. We have added the detailed information of the implantable electrodes in the revised manuscript including the thickness of each component and images of the implantable electrodes.

Fig. R11 (added as Supplementary Fig. 17) | Electrodes with the adhesive and non-adhesive interfaces. **a**, Schematic illustrations for electrodes with the adhesive and the non-adhesive interfaces. **b,c**, Photographs of electrodes with the adhesive interface (**b**) and the non-adhesive interface (**c**) implanted to the rat epicardial surface. White dotted lines in photographs indicate the boundary of implants.

On Page 7, “**Preparation of implantable electrodes.** To prepare the implantable electrodes, the gold electrodes (thickness, 50 μm) were integrated between the polyurethane layer (thickness, 100 μm) and the adhesive or non-adhesive layer (thickness, 100 μm) (Supplementary Fig. 17a). The surface of the gold electrode was treated with oxygen plasma for 3 min (30 W power, Harrick Plasma) to activate the surface functionalization, followed by immersing in cysteamine hydrochloride solution (50 mM in deionized water) for 1 h at room temperature. After the functionalization, the gold electrode was thoroughly washed with deionized water and dried with nitrogen flow. The functionalized gold electrode was cut into 2-mm diameter circles and placed on the adhesive hydrogel (two electrodes per implant). An electrode lead wire (AS633, Cooner Wire) was connected to the gold electrodes and the polyurethane insulation layer (HydroThane, AdvanSource Biomaterials) was introduced to the gold electrodes. The assembled implant was thoroughly dried under airflow and a vacuum desiccator to prepare the adhesive implantable electrodes. To prepare the non-adhesive implantable electrodes, the adhesive implantable electrodes were immersed in a sterile PBS bath overnight. All samples were prepared in an aseptic manner and were further disinfected under UV for 1 h before use.”

We greatly appreciate the reviewer’s time and effort to provide insightful comments and suggestions. We hope that our revised manuscript and responses address all of your concerns about the work.

Reviewer Reports on the First Revision:

Referees' comments:

Referee #1 (Remarks to the Author):

The authors have addressed my concerns with the additional evaluation in a wider number of preclinical models and demonstrated the anti-fibrotic approach.

Referee #3 (Remarks to the Author):

The authors have provided additional convincing data, including a longer time of in vivo studies in rat models, different animal models, different adhesive compositions, and adhesive implant with sutures, among others. All of my prior concerns have been satisfactorily addressed.

Minor issues: In supplementary Fig. 8a, the x axis (Stretch) lacked a unit. The "Mechanical characterizations" section lacked a replicate number for each test.

Referee #4 (Remarks to the Author):

In this report, the authors used various assays, including in vitro protein adsorption, multiplex Luminex, quantitative PCR, immunofluorescence analysis, and RNA sequencing to validate the hypothesis that fibrous capsule formation around adhesive implants can be reduced by limiting the inflammatory response (neutrophils and monocytes) that drives capsule formation around non-adhesive implants.

1. The foreign body reaction that normally activates inflammation was fully characterized in response to both types of implants. Overall, it appears the adhesive implants resulted in reduced infiltration of inflammatory cells overall with fewer fibroblasts (aSMA/coll), T cells, neutrophils and macrophages (both pro- and anti-inflammatory macs) relative to non-adhesive interfaces (Fig 3g-i). Surprisingly, however, the adhesive implants showed increased inflammatory cytokine production with greater IL-1/MCP1 on day 1 and increased IL-12 and GM-CSF on day 7 (Fig 4b). In general, the authors could do a much better job describing these changes as they suggest the adhesive implants subsequently have lower cytokine production than the non-adhesive, which is not clear at all to me in the figure (4b). The authors need to add statistics to the Luminex data in figure 4b and modify the associated interpretation. To me, it seems the adhesive interfaces induce a more rapid and intense inflammatory than the non-adhesive implants, which is contrary to the authors' conclusions.

2. Also, why are the gene expression analyses included in figures 4 c-f not consistent across time points? Some genes are included at some time points but not others. Again, the data for Nos2 at day 1 and 3 suggest a more robust and early pro-inflammatory neutrophil/macrophage response is induced by

the adhesive interface. While the data at day 7 showed a clear reduction in all cytokine mRNAs, perhaps this simply reflects a more rapid resolution of the inflammatory response that peaked earlier with the adhesive implants? Again, I'm not sure these adhesive implants are overall less inflammatory than the non-adhesive implants, so I question the author's conclusions on this front.

3. For me, some of the most impactful data are shared in figure 5. Sadly, single time points are shared for each model (day 28 for mice and day 7 for the pig). Given the striking difference observed between the adhesive and non-adhesive interfaces in the mouse, it would be helpful to evaluate the inflammatory response at one earlier time point, minimally. For this reviewer, the strikingly more severe late-stage fibrosis seen here with non-adhesive interface suggests the "earlier" pro-inflammatory response observed with adhesive interface may in fact be anti-fibrotic. It's striking that there is virtually no inflammation or fibrosis with the adhesive interface at this time point. But what about earlier time points? Perhaps the inflammation is again resolving much faster in this group, leading to less fibrosis later? The data from the pig model suggests this might be the case since the differences here was much less dramatic on day 7, than the day 28 timepoint in mice. The native tissue in the pig also looks more similar to the non-adhesive interface so it's more difficult to interpret these data.

Overall, based on my interpretation of the data provided, it's difficult for me to support the author's overall conclusion that adhesive implants lead to less fibrous capsule formation by minimizing the overall inflammatory response. There is simply too much data provided that supports the opposite conclusion. Perhaps instead, the character of the rapid inflammatory response induced by the adhesive interface leads to the development of an anti-fibrotic response? A more detailed analysis of the data generated in figure 5 would be needed to support this alternative conclusion.

Author Rebuttals to First Revision:

Referee #1 (Remarks to the Author):

Comment. The authors have addressed my concerns with the additional evaluation in a wider number of preclinical models and demonstrated the anti-fibrotic approach.

Response. We are pleased to learn that the revised manuscript has addressed the reviewer's comments and concerns in full. The manuscript has been greatly improved through the revision and we greatly appreciate the reviewer's time and effort to provide insightful comments and suggestions.

Referee #3 (Remarks to the Author):

Comment. The authors have provided additional convincing data, including a longer time of in vivo studies in rat models, different animal models, different adhesive compositions, and adhesive implant with sutures, among others. All of my prior concerns have been satisfactorily addressed. Minor issues: In supplementary Fig. 8a, the x axis (Stretch) lacked a unit. The "Mechanical characterizations" section lacked a replicate number for each test.

Response. We are pleased to learn that the revised manuscript has addressed the reviewer's comments and concerns in full.

Regarding the units for Stretch in Supplementary Fig. 8a, we would like to clarify that Stretch is unitless, and therefore, there was no unit indicated in the x-axis of Fig. 8a. We also clarified the number of replicates for each test in the Method "Mechanical Characterization" section in the revised manuscript as below:

On Page 6, "All mechanical characterizations were performed 3 times based on independently prepared samples."

Referee #4 (Remarks to the Author):

General Comment. In this report, the authors used various assays, including in vitro protein adsorption, multiplex Luminex, quantitative PCR, immunofluorescence analysis, and RNA sequencing to validate the hypothesis that fibrous capsule formation around adhesive implants can be reduced by limiting the inflammatory response (neutrophils and monocytes) that drives capsule formation around non-adhesive implants.

Comment. Thank you for your insightful feedback on our work. To fully address the reviewer's comments and concerns, we have revised the manuscript by revising 2 Main Figures and adding 1 Supplementary Figure, 1 Supplementary Table, analyses, and clarifications. The newly added text in the revised manuscript is highlighted in blue color. In particular, to fully address the reviewer's suggestions and concerns, an immunology expert has performed thorough immunological analyses and interpretations in the revised manuscript as a co-author.

Comment 1. The foreign body reaction that normally activates inflammation was fully characterized in response to both types of implants. Overall, it appears the adhesive implants resulted in reduced infiltration of inflammatory cells overall with fewer fibroblasts (α SMA/coll), T cells, neutrophils and macrophages (both pro- and anti-inflammatory macs) relative to non-adhesive interfaces (Fig 3g-i). Surprisingly, however, the adhesive implants showed increased inflammatory cytokine production with greater IL-1/MCP1 on day 1 and increased IL-12 and GM-CSF on day 7 (Fig 4b). In general, the authors could do a much better job describing these changes as they suggest the adhesive implants subsequently have lower cytokine production than the non-adhesive, which is not clear at all to me in the figure (4b). The authors need to add statistics to the Luminex data in figure 4b and modify the associated interpretation. To me, it seems the adhesive interfaces induce a more rapid and intense inflammatory than the non-adhesive implants, which is contrary to the authors conclusions.

Response 1. Thank you for your insightful comments. We would like to emphasize that our conclusion and supporting data are fully consistent with the reviewer's comment: the adhesive implant-tissue interface can mitigate fibrous capsule formation due to the reduced infiltration of inflammatory cells and subsequent reduction in the formation of fibrous capsules in the adhesive implant-tissue interface compared to the non-adhesive implant-tissue interface.

As suggested by the reviewer, we thoroughly re-analyzed the immune cell-related genes and cytokines to better understand the immunological aspects of the adhesive and non-adhesive implant-tissue interfaces as described below in detail:

1) Re-analysis of immunofluorescence data (see Fig. R1 below)

We re-analyzed our immunofluorescence data to better quantify the numbers of fibroblasts (α SMA), neutrophils (neutrophil elastase), macrophages (CD68 for pan-macrophages; iNOS and vimentin for pro-inflammatory macrophages; CD206 for anti-inflammatory macrophages), and T-cells (CD3) at the adhesive and non-adhesive implant-tissue interfaces on days 3, 7, and 14 post-implantation. The analysis showed significantly fewer fibroblasts, neutrophils, macrophages, and T-cells at the adhesive implant-tissue interface than at the non-adhesive implant-tissue interface at all time points. These results validate that there are significantly fewer infiltrating inflammatory cells at adhesive implant-tissue interface than the non-adhesive implant-tissue interface.

2) Re-analysis of q-PCR and Luminex multiplex assay data (see Fig. R2)

Based on the re-analysis of immunofluorescence data in 1), correspondingly, we also re-analyzed q-PCR and Luminex multiplex assay data focusing on days 3 and 7 post-implantation to better understand immune responses at the adhesive and non-adhesive implant-tissue interfaces. We excluded an acute time point (day 1 post-implantation) to avoid potential influences from the surgery. The analysis showed that:

- i) On day 3 post-implantation, whereas the levels of most select immune gene transcripts are similar or significantly lower in the adhesive as compared to the non-adhesive implant-tissue interface, the level of *Nos2* is significantly higher in the adhesive than the non-adhesive implant-tissue interface. The higher level of *Nos2* expression is in agreement with the higher levels of inflammatory cytokines (G-CSF, IL-12p70) in the adhesive than the non-adhesive implant-tissue interface on day 3 post-implantation.
- ii) On day 7 post-implantation, the adhesive implant-tissue interface exhibits a significantly lower expression of all immune cell-related genes, including *Nos2*, compared to the non-adhesive implant-tissue interface, consistent with the reduction in inflammatory cytokines in the adhesive implant-tissue interface on day 7 post-implantation as compared to day 3 post-implantation.

3) Investigation of the source of *Nos2* (see Fig. R3 below)

Based on the re-analysis of q-PCR and Luminex multiplex assay data in 2), we further investigated the source of significantly higher *Nos2* and associated inflammatory cytokines on day 3 post-implantation. For this purpose, we performed double immunofluorescence staining for iNOS/Neutrophil elastase and iNOS/CD68 in the adhesive and non-adhesive implant-tissue interfaces on day 3 post-implantation. The analysis showed that:

- i) The adhesive implant-tissue interface showed a significantly higher number of iNOS+ neutrophils rather than iNOS+ macrophages on day 3 post-implantation.
- ii) The non-adhesive implant-tissue interface showed similar numbers of iNOS+ neutrophils and iNOS+ macrophages on day 3 post-implantation, contrary to the adhesive implant-tissue interface.

These results indicate that the adhesive implant-tissue interface favors an iNOS-producing neutrophil subset in the acute time points (day 3) after implantation. Based on the outcome of our analyses, we revised our conclusion regarding the immunological aspect as the following: the adhesive implant-tissue interface seems to induce a more robust pro-inflammatory neutrophil response than the non-adhesive implant-tissue interface at the acute time points (day 3 post-implantation), which is rapidly resolved by day 7 post-implantation.

We also wish to note that our revised conclusion is in agreement with the reviewer's comment that "*it seems the adhesive interfaces induce a more rapid and intense inflammatory than the non-adhesive implants.*"

To reflect these points, we have added the following paragraphs and figures in the revised manuscript. Also, the statistical analysis details of the Luminex multiplex assay data are added as Supplementary Table 1 in the revised manuscript accordingly.

On Page 3, "To investigate the infiltration of immune cells into the implant-tissue interface, we performed immunofluorescence staining for fibroblasts (α SMA), neutrophils (neutrophil elastase), macrophages (CD68 for pan-macrophages; iNOS and vimentin for pro-inflammatory macrophages; CD206 for anti-inflammatory macrophages), and T-cells (CD3) on days 3, 7, and

14 post-implantation (**Fig. 3a-f**). Quantification of cell numbers in the collagenous layer at the implant-tissue interface over a representative width of 500µm from the immunofluorescence images shows significantly fewer fibroblasts, neutrophils, macrophages, and T-cells at the adhesive implant-tissue interface than at the non-adhesive implant-tissue interface at all time points (**Fig. 3g-i**).

To further delineate the immune response at the implant-tissue interface, we profile immune cell-related genes and cytokines using qPCR analysis and Luminex quantification, respectively (**Fig. 4**). On day 3 post-implantation, whereas the levels of most select immune gene transcripts are similar or significantly lower in the adhesive as compared to the non-adhesive implant-tissue interface, the level of *Nos2* is significantly higher in the adhesive than the non-adhesive implant-tissue interface (**Fig. 4b**). The higher level of *Nos2* expression is in agreement with the higher levels of inflammatory cytokines (G-CSF, IL-12p70) in the adhesive than the non-adhesive implant-tissue interface on day 3 post-implantation (**Fig. 4d** and **Supplementary Table 1**).”

On Page 4, “To investigate the source of *Nos2* on day 3 post-implantation, we performed double immunofluorescence staining for iNOS/Neutrophil elastase and iNOS/CD68 (**Supplementary Fig. 11**). The immunofluorescence staining for the adhesive implant-tissue interface reveals a significantly higher number of iNOS+ neutrophils rather than iNOS+ macrophages on day 3 post-implantation ($P \leq 0.01$) (**Supplementary Fig. 11b**). In contrast, the non-adhesive implant-tissue interface has similar numbers of iNOS+ neutrophils and iNOS+ macrophages on day 3 post-implantation ($P = 0.82$) (**Supplementary Fig. 11d**). This result indicates that the adhesive implant-tissue interface favors an iNOS-producing neutrophil subset on day 3 post-implantation³¹.

By day 7 post-implantation, the adhesive implant-tissue interface exhibits a significantly lower expression of all immune cell-related genes, including *Nos2*, compared to the non-adhesive implant-tissue interface (**Fig. 4c**), consistent with the reduction in inflammatory cytokines in the adhesive implant-tissue interface on day 7 post-implantation as compared to day 3 post-implantation (**Fig. 4d** and **Supplementary Table 1**). Thus, the adhesive implant-tissue interface seems to induce a more robust pro-inflammatory neutrophil response than the non-adhesive implant-tissue interface on day 3 post-implantation, which is rapidly resolved by day 7 post-implantation.”

Fig. R1 | Immunofluorescence analysis of the adhesive and non-adhesive implant-tissue interfaces at different time points. a,c,e, Representative immunofluorescence images of the non-adhesive implant collected on day 3 (a), day 7 (c), and day 14 (e) post-implantation on the abdominal wall. b,d,f, Representative immunofluorescence images of the adhesive implant collected on day 3 (b), day 7 (d), and day 14 (f) post-implantation on the abdominal wall. In immunofluorescence images, cell nuclei are stained

with 4',6-diamidino-2-phenylindole (DAPI, blue); green fluorescence corresponds to the expression of fibroblast (α SMA), neutrophil (Neutrophil elastase), and macrophage (CD68, Vimentin, CD206, iNOS); red fluorescence corresponds to the expression of T cell (CD3). * in images indicates the implant; white dotted lines in images indicate the implant-tissue interface; yellow dotted lines in images indicate either the mesothelium-fibrous capsule interface (non-adhesive implant) or the mesothelium-skeletal muscle interface (adhesive implant). **g-i**, Quantification of cell numbers in the collagenous layer at the implant-tissue interface over a representative width of 500 μ m from the immunofluorescence images on day 3 (g), day 7 (h), and day 14 (i) post-implantation. Values in **g-i** represent the mean and the standard deviation ($n = 3$ implants; independent biological replicates). Statistical significance and P values are determined by two-sided unpaired t -tests; ns, not significant; * $P < 0.05$; ** $P \leq 0.01$; *** $P \leq 0.001$; **** $P < 0.0001$.

Fig. R2 | Q-PCR and Luminex analysis of the adhesive and non-adhesive implant-tissue interfaces. **a**, Genes and cytokines relevant to each cell type in the q-PCR and Luminex studies. **b-c**, Normalized gene expression of immune cell-related markers for the non-adhesive and the adhesive implant-tissue interface collected on day 3 (b) and day 7 (c) post-implantations on the abdominal wall. **d**, Heatmap of immune cell-

related cytokines measured with Luminex assay of the non-adhesive and the adhesive implant-tissue interface collected on days 3 and 7 post-implantations on the abdominal wall. Values in **b,c** represent the mean and the standard deviation ($n = 9$ implants; independent biological replicates). Statistical significance and P values are determined by two-sided unpaired t -tests; ns, not significant; $*P < 0.05$; $**P \leq 0.01$; $***P \leq 0.001$; $****P < 0.0001$.

Fig. R3 | Immunofluorescence analysis of iNOS+ cells at the implant-tissue interface. **a**, Representative immunofluorescence images at the adhesive implant-tissue interface on day 3 post-implantation to the abdominal wall. **b**, Quantification of iNOS+/Neutrophil Elastase+ and iNOS+/CD68+ cells per unit area on day 3 post-implantation for adhesive implant-tissue interface. **c**, Representative immunofluorescence images at the non-adhesive implant-tissue interface on day 3 post-implantation to the abdominal wall. **d**, Quantification of iNOS+/Neutrophil Elastase+ and iNOS+/CD68+ cells per unit area on day 3 post-implantation for non-adhesive implant-tissue interface. In immunofluorescence images, cell nuclei are stained with 4',6-diamidino-2-phenylindole (DAPI, blue); green fluorescence corresponds to the expression of macrophage (CD68) and neutrophil (Neutrophil Elastase); red fluorescence corresponds to the expression of iNOS. * in images indicates the implant; white dotted lines in images indicate the implant-tissue interface. Values in **b** and **d** represent the mean and the standard deviation ($n = 3$ implants; independent biological replicates). Statistical significance and P values are determined by two-sided unpaired t -tests; ns, not significant; $**P \leq 0.01$.

Supplementary Table 1 | Luminex multiplex assays for protein profiling of rat abdominal wall tissues. Data were normalized per total amount of proteins extracted and are represented as the mean \pm standard deviation ($n = 3$ tissue lysates; independent biological replicates). Statistical significance and P values are determined by two-sided unpaired t -tests; $*P < 0.05$ (vs. non-adhesive interface for the same time point).

Molecule (pg mg ⁻¹ protein)	Non-adhesive interface	Adhesive interface
---	------------------------	--------------------

	Day 3	Day 7	Day 3	Day 7
IL-1 α	6.87 \pm 2.61	10.47 \pm 5.38	3.87 \pm 0.45	8.75 \pm 5.08
IL-1 β	107.65 \pm 129.51	125.97 \pm 130.24	20.95 \pm 4	26.77 \pm 8.04
IL-6	31.48 \pm 45.88	50.44 \pm 60	15.31 \pm 11.07	18.74 \pm 6.33
IL-12P70	1.05 \pm 0.23	1.18 \pm 0.32	1.74 \pm 0.28*	1.7 \pm 0.35
IL-17	3.18 \pm 0.34	4.04 \pm 1.2	3.22 \pm 0.47	3.74 \pm 0.2
IL-18	394.21 \pm 20.14	430.18 \pm 111.11	192.43 \pm 28.83*	145.5 \pm 12.14*
TNF- α	0.58 \pm 0.02	0.98 \pm 0.4	0.69 \pm 0.07	0.65 \pm 0.11
MCP-1	53.77 \pm 44.43	64.64 \pm 28.67	47.47 \pm 16.12	34.17 \pm 2.23
G-CSF	0.2 \pm 0.05	0.25 \pm 0.03	0.36 \pm 0.06*	0.34 \pm 0.06
IL-4	12.62 \pm 5.8	12.61 \pm 1.71	17.41 \pm 3.46	13.13 \pm 1.61
IL-10	4.64 \pm 2.63	9.93 \pm 11.36	3.03 \pm 0.35	2.31 \pm 0.25
CCL3	17.34 \pm 3.56	77.24 \pm 73.93	9.21 \pm 1.15*	20.74 \pm 14.28
CCL5	10.15 \pm 1.49	69.24 \pm 31.03	5.88 \pm 0.84*	38.57 \pm 25.39

Comment 2. Also, why are the gene expression analyses included in figures 4 c-f not consistent across time points? Some genes are included at some time points but not others. Again, the data for Nos2 at day 1 and 3 suggest a more robust and early pro-inflammatory neutrophil/macrophage response is induced by the adhesive interface. While the data at day 7 showed a clear reduction in all cytokine mRNAs, perhaps this simply reflects a more rapid resolution of the inflammatory response that peaked earlier with the adhesive implants? Again, I'm not sure these adhesive implants are overall less inflammatory than the non-adhesive implants, so I question the author's conclusions on this front.

Response 2. Thank you for your insightful comment. The genes for q-PCR analysis in **Fig. 4** have been revised to be consistent across all time points in the revised manuscript (**Fig. R2**). For other comments, please refer to **Response 1** above.

Comment 3. For me, some of the most impactful data are shared in figure 5. Sadly, single time points are shared for each model (day 28 for mice and day 7 for the pig). Given the striking difference observed between the adhesive and non-adhesive interfaces in the mouse, it would be helpful to evaluate the inflammatory response at one earlier time point, minimally. For this reviewer, the strikingly more severe late-stage fibrosis seen here with non-adhesive interface suggests the "earlier" pro-inflammatory response observed with adhesive interface may in fact be anti-fibrotic. It's striking that there is virtually no inflammation or fibrosis with the adhesive interface at this time point. But what about earlier time points? Perhaps the inflammation is again resolving much faster in this group, leading to less fibrosis later? The data from the pig model suggests this might be the case since the differences here was much less dramatic on day 7, than the day 28

timepoint in mice. The native tissue in the pig also looks more similar to the non-adhesive interface so it's more difficult to interpret these data.

Overall, based on my interpretation of the data provided, it's difficult for me to support the author's overall conclusion that adhesive implants lead to less fibrous capsule formation by minimizing the overall inflammatory response. There is simply too much data provided that supports the opposite conclusion. Perhaps instead, the character of the rapid inflammatory response induced by the adhesive interface leads to the development of an anti-fibrotic response? A more detailed analysis of the data generated in figure 5 would be needed to support this alternative conclusion.

Response 3. We would like to thank the reviewer for his/her insightful comments. Our revised manuscript is consistent with the reviewer's interpretation of the data. We would also like to clarify that the detailed in-depth experiments and analyses of the proposed hypothesis were provided for a wide range of time points (day 3 to day 84 post-implantation) in rat models based on histopathology, immunofluorescence, q-PCR, Luminex assay, and RNA-seq (**Figs. R1-R3**). The main purpose of Fig. 5 is to demonstrate the anti-fibrotic interfaces in diverse animal models (C57BL/6 mice, HuCD34-NCG humanized mice, pigs) rather than duplicating the earlier analyses.

We also wish to emphasize that we have the results on C57BL/6 mice at an earlier time point, day 14 post-implantation, which are consistent with the results on day 28 post-implantation (see **Fig. R4** below). Additionally, in the pig model, we validate the adhesive anti-fibrotic implant-tissue interface on day 7 post-implantation not only in the abdominal wall but also in the small intestine (see **Fig. R5** below).

We greatly appreciate the reviewer's time and effort to provide insightful comments and suggestions. We hope that our revised manuscript and responses sufficiently address the reviewer's concerns. Thank you very much.

Fig. R4 | Adhesive anti-fibrotic interfaces in C57BL/6 mice. a,b, Representative histology images stained with Masson's trichrome (MTS) and hematoxylin and eosin (H&E) for native tissue (left), adhesive implant (middle), and non-adhesive implant (right) collected on day 14 (a) and day 28 (b) post-implantation

in C57BL/6 mice. Black dotted lines in images indicate the implant-tissue interface; yellow dotted lines in images indicate the fibrous capsule-tissue interface.

Fig. R5 | Adhesive anti-fibrotic interfaces in porcine model. a, b Representative histology images stained with Masson's trichrome (MTS) and hematoxylin and eosin (H&E) for native tissue (left), adhesive implant (middle), and non-adhesive implant (right) collected on 7 days post-implantation to the small intestine (a) and abdominal wall (b). Black dotted lines in images indicate the implant-tissue interface; yellow dotted lines in images indicate the fibrous capsule-tissue interface.

Reviewer Reports on the Second Revision:

Referees' comments:

Referee #4 (Remarks to the Author):

I'm pleased with how the author's carefully addressed my concerns regarding the early versus late inflammatory response. The revised manuscript is greatly improved and more aligned with the data.

Author Rebuttals to Second Revision:

Referee #4 (Remarks to the Author):

Comment. I'm pleased with how the author's carefully addressed my concerns regarding the early versus late inflammatory response. The revised manuscript is greatly improved and more aligned with the data.

Response. We are pleased to learn that the revised manuscript has addressed the reviewer's comments and concerns in full. The manuscript has been greatly improved through the revision and we greatly appreciate the reviewer's time and effort to provide insightful comments and suggestions.